# Exogenous Sugar Alleviates Salt Stress in Cucumber Seedlings by Regulating the Antioxidant System and Hormone Signaling

**DOI:** 10.3390/cimb47090754

**Published:** 2025-09-12

**Authors:** Guangchao Yu, Zhipeng Wang, Ming Wei, Lian Jia, Yue Qu, Yingyi Jiang

**Affiliations:** 1College of Chemistry and Life Sciences, Anshan Normal University, Anshan 114007, China; wangzp1326@gmail.com (Z.W.); wm050820@163.com (M.W.); jl_58@163.com (L.J.); quyue199209@163.com (Y.Q.); jiangdi031507@163.com (Y.J.); 2Liaoning Key Laboratory of Development and Utilization for Natural Products Active Molecules, Anshan Normal University, Anshan 114007, China

**Keywords:** cucumber, salt stress, exogenous sugar, antioxidant enzymes, hormone signaling, transcription factors

## Abstract

This study explored the regulatory effects of exogenous glucose (Glu) and sucrose (Suc) on the growth performance and physiological mechanisms of cucumber seedlings under salt stress. Using two cucumber cultivars as experimental materials, pot experiments demonstrated that salt stress significantly suppressed seedling growth, decreased chlorophyll content, and triggered oxidative damage. However, pretreatment with exogenous sugars effectively mitigated these adverse effects by maintaining photosynthetic efficiency, enhancing the activities of key antioxidant enzymes—superoxide dismutase (SOD), peroxidase (POD), catalase (CAT), and ascorbate peroxidase (APX)—and reducing the accumulation of reactive oxygen species (ROS) and membrane lipid peroxidation. Transcriptomic analysis revealed that the two sugars differentially modulated antioxidant pathways and transcription factor networks to synergistically enhance salt tolerance. Specifically, sucrose preferentially activated POD, whereas glucose specifically induced APX and RbohD. Furthermore, glucose upregulated NAC and ERF family genes, while sucrose suppressed certain WRKY members. Both sugars contributed to the restoration of auxin and abscisic acid (ABA) signaling pathways. This study provides a theoretical foundation for the role of sugar signaling in enhancing crop resistance to abiotic stress.

## 1. Introduction

Plants in natural environments are frequently exposed to various abiotic stresses, such as salt stress and low-temperature stress, among which salt stress exerts a particularly profound impact on plant growth and development [1,2]. Salt stress adversely affects plant morphogenesis, including reductions in root length, stem length, fresh weight, and dry weight. This inhibition primarily results from the reduced water absorption capacity of the root system under high soil salinity, which induces osmotic stress and ultimately leads to decreased leaf relative water content and delayed plant growth [3]. Under high-salt conditions, excessive accumulation of reactive oxygen species (ROS), such as H_2_O_2_ and O_2_^−^, occurs within plant cells. This leads to oxidative damage, compromising cell membrane integrity (as indicated by elevated MDA levels) and damaging macromolecules, such as proteins and DNA. To counteract oxidative stress, plants activate their antioxidant defense systems, which involve the upregulation of antioxidant enzymes (SOD, POD, CAT, APX) and the synthesis of non-enzymatic antioxidants (such as AsA and GSH), thereby maintaining cellular ROS homeostasis [4,5]. Research has found that salt stress activates the soybean transcription factor GmNTL1 (at the key site of Cys-247) through H_2_O_2_-mediated oxidative modification, causing it to transfer from the endoplasmic reticulum to the nucleus. This then initiates the expression of genes related to ROS generation (*GmRbohB*) and ion homeostasis (GmSALT3/GmNHX1), forming a positive feedback loop to enhance salt tolerance [6]. Zhang et al. found that the ectopic expression of six key antioxidant enzyme genes (*GOX*, *PrxR*, *Trx*, *APX*, *MDHAR*, and *DHAR3*) in the beet M14 line effectively regulated redox metabolism (such as H_2_O_2_, MDA, AsA, and GSH levels), significantly enhancing the salt stress tolerance of transgenic plants [7], providing valuable genetic resources for crop salt-tolerance breeding. However, when salt stress surpasses the adaptive capacity of plants, growth and yield are significantly impaired. This issue is especially pronounced in crops like cucumber, posing a direct threat to global food security.

Cucumber (*Cucumis sativus* L.), a member of the Cucurbitaceae family, is an economically significant crop. It has specific environmental requirements and thrives best in neutral to slightly acidic soils, yet is highly susceptible to salt–alkali stress [8]. Soil salinization reduces the population of soil fauna and microorganisms, decreases plant chlorophyll content, and suppresses both photosynthetic and transpiration rates. These physiological impairments directly lead to a range of morphological and physiological alterations, such as root rot, wilting, leaf chlorosis, and reduced fruiting, which severely compromise cucumber yield and product quality [9]. The germination and seedling stages are particularly sensitive to salt stress and are critical for assessing cucumber salt tolerance. Salt stress has been shown to significantly inhibit seed germination and negatively affect various germination-related parameters [10,11]. Research has demonstrated that under salt stress conditions, cucumber seedlings rapidly elevate endogenous H_2_O_2_ levels through the activation of plasma membrane NADPH oxidases (RBOHs) and the corresponding gene expression, such as that of the CsRboh gene. This response is accompanied by increased activity of glucose-6-phosphate dehydrogenase (G6PDH), which provides the necessary substrates for enzymatic reactions. With prolonged exposure to salt stress, the antioxidant enzyme system is subsequently activated to maintain reactive oxygen species (ROS) homeostasis. These findings indicate that the NADPH oxidase-mediated H_2_O_2_ burst functions as an early and crucial response mechanism that enables cucumber seedlings to adapt to salt stress [12]. Recent studies have explored the application of exogenous substances to alleviate salt stress-induced damage in cucumbers, yielding promising outcomes. Abdel-Farid et al. reported that under 200 mM NaCl stress, cucumber seed germination was delayed, with significant reductions observed in both germination percentage and seedling stem length. Concurrently, the levels of phenolic and flavonoid compounds increased, and metabolomic analysis revealed elevated levels of biologically active secondary metabolites [13]. Further studies have demonstrated that increasing salinity levels result in significant declines in growth parameters and chlorophyll content (SPAD values) of cucumber seedlings, accompanied by suppressed photosynthetic performance. However, antioxidant enzyme activities, along with proline and sucrose concentrations in leaves, were found to increase. Notably, hydrogen sulfide treatment has been shown to effectively alleviate salt stress-induced physiological damage in cucumber seedlings [8]. Zhang et al. found that melatonin application under salt stress conditions can enhance cucumber seed germination. Proteomic analysis revealed that melatonin significantly modulates key metabolic pathways, including ribosome biogenesis, lipid metabolism, carbohydrate metabolism, and the degradation of storage proteins. Moreover, the expression levels of stress-related proteins, as well as those involved in glycolysis, the tricarboxylic acid (TCA) cycle, the glyoxylate cycle, and ATP synthesis, were markedly upregulated [14].

Sugar plays an essential role throughout the plant life cycle, serving not only as an energy source and a key component in cell wall biosynthesis but also functioning as a primary signaling molecule with hormone-like regulatory effects on plant growth, development, and gene expression [15]. Soluble sugars occupy a central position in cellular redox regulation by balancing the processes of photosynthesis, respiration, and β-oxidation of fatty acids [16]. In recent years, the application of exogenous substances has emerged as a promising strategy to enhance plant stress tolerance. Among these, exogenous sugars—such as glucose and sucrose—have been demonstrated to mitigate abiotic stress through multiple physiological and molecular mechanisms. Under salt stress conditions, exogenous sucrose has been shown to elevate the levels of glucose and fructose in salt-sensitive rice varieties, thereby improving their salt tolerance [17]. Boriboonkase’s research revealed that rice seeds pretreated with sucrose exhibited enhanced resistance under high-salt stress (342 mmol·L^−1^ NaCl), highlighting the critical regulatory role of sugar signaling in plant responses to adverse environmental conditions [18]. Exogenous sugars can facilitate osmotic adjustment by promoting the accumulation of osmoprotectants, such as soluble proteins and soluble sugars, thus maintaining cellular water homeostasis. The accumulation of soluble sugars has been found to play a crucial role in osmoregulation under both salt and drought stress conditions [19,20]. Studies have indicated that under salt stress, exogenous sugar significantly increases the content of soluble sugars and soluble proteins in asparagus, thereby enhancing its osmotic adjustment capacity [21]. Furthermore, exogenous sugar can strengthen the ROS scavenging system, leading to reduced levels of MDA and H_2_O_2_ and alleviating oxidative damage [16]. For instance, treatment with 50 mM exogenous sucrose was shown to significantly increase the activities of SOD, POD, and CAT in salt-stressed cucumber seedlings while decreasing H_2_O_2_ and MDA content, thereby mitigating membrane lipid peroxidation [22]. Research has also demonstrated that exogenous sugars, including glucose and sucrose, help maintain redox homeostasis by modulating NADPH oxidase (RBOH) activity and antioxidant enzyme expression, ultimately reducing ROS accumulation [23]. Additionally, studies have found that exogenous sucrose, particularly when combined with salicylic acid (SA), significantly enhances the activities of SOD and CAT in maize, lowers H_2_O_2_ and MDA levels, and improves drought tolerance [24]. Moreover, exogenous sugars, such as sucrose and trehalose, contribute to osmotic balance by promoting the accumulation of soluble sugars and proline while simultaneously reducing ROS-induced cellular damage [25].

As a signaling molecule, sugar may participate in the transduction of hormonal signals (ABA, ET, AUX), thereby coordinating the expression of stress-responsive genes. The ABA receptor PYL/RCAR interacts with PP2C phosphatases (e.g., ABI1, ABI2), thereby releasing the inhibition of SnRK2 kinases and activating downstream antioxidant genes (e.g., SOD, CAT) as well as osmotic adjustment genes (e.g., NHX1) [26]. NHX1/2 contributes to vacuolar Na^+^ homeostasis, which reduces ROS accumulation under salt stress, and its expression is co-regulated by both ABA and sugar signaling pathways [27]. Studies have demonstrated that ERF transcription factors (e.g., ERF1, ERF6) integrate ethylene (ET) and sugar signals to activate antioxidant genes (e.g., APX2, GST) and osmotic adjustment-related genes [28]. Sugar-induced members of the ERF-VII subfamily enhance plant salt tolerance through the regulation of H_2_O_2_ metabolism [29]. Furthermore, ARF transcription factors (e.g., ARF7/19) and SAUR genes are modulated by sugar signaling, promoting root growth via the regulation of IAA distribution and indirectly enhancing ROS scavenging capacity [30]. Sucrose promotes YUCCA (IAA synthase) expression through PIF4-mediated regulation, thereby facilitating ARF-dependent activation of antioxidant genes [31].

Transcription factors (TFs) serve as central regulators in plant responses to salt stress, orchestrating physiological and molecular adaptive mechanisms through the activation or repression of downstream target genes. RNA-seq analysis has revealed that under salt stress conditions, multiple TF families are activated in citrus, including WRKY, NAC, MYB, AP2/ERF, bZIP, GATA, bHLH, ZFP, SPL, CBF, and CAMTA [32]. Research has demonstrated that the NAC domain-containing transcription factors GmNAC06 from soybean and AvNAC030 from kiwifruit enhance plant salt tolerance [33,34]. The peanut MYB-related transcription factor AhMYB30 improves freezing and salt tolerance in transgenic Arabidopsis via the DREB/CBF and ABA signaling pathways [35]. Overexpression of SaR2R3-MYB15 in transgenic Astragalus complanatus plants increases antioxidant enzyme activity and proline accumulation, reduces malondialdehyde (MDA) content, and thereby enhances salt stress tolerance [36]. Overexpression of the DcERF3 gene promotes lateral root development in Arabidopsis and improves its tolerance to salt stress [37]. Transgenic overexpression of the WRKY transcription factor McWRKY57 in Mentha canadensis enhances drought tolerance in Arabidopsis [38]. Oxidation of the C2H2-type zinc finger protein ZFP36 upregulates both the expression and activity of genes encoding protective antioxidant enzymes [39]. In Suaeda glauca, enhanced expression of ABA metabolism-related genes such as AtABA1, AtABA3, AtNCED3, and AtAAO3 contributes to improved salt tolerance [40].

With advances in high-throughput sequencing technology and comparative genomics, transcriptome sequencing (RNA-seq) has emerged as a powerful tool for elucidating the molecular mechanisms underlying plant responses to adverse environmental conditions. The completion of the cucumber genome sequence has provided a solid foundation for investigating the genetic basis of its salt tolerance. However, studies exploring how exogenous sugars enhance cucumber salt tolerance through transcriptome reprogramming remain limited, particularly regarding the molecular interactions between sugar signaling and hormonal pathways, which are still poorly understood. In this study, cucumber seedlings were used as experimental materials. Glucose and sucrose were applied exogenously, and the ameliorative effects of these sugars on salt stress were investigated through physiological measurements combined with transcriptomic analysis. The specific research hypotheses are as follows. (1) Exogenous sugars alleviate salt stress-induced growth inhibition in cucumber seedlings. (2) Exogenous sugars activate the antioxidant defense system, maintain osmotic homeostasis, and regulate hormonal balance. (3) Exogenous sugars trigger the differential expression of salt stress-responsive genes, including those encoding antioxidant enzymes, osmoregulatory proteins, and components of hormone signal transduction pathways, thereby enhancing salt tolerance through transcriptome reprogramming.

## 2. Materials and Methods

### 2.1. Plant Cultivation and Treatment

The cucumber varieties used in the experiment were Xintaimici (salt-tolerant cucumber variety) and Jinyou1 (salt-sensitive cucumber variety). Plump and uniform seeds were selected and soaked in water for about 6 h and then germinated in the dark at 28 °C. After the seeds germinated, uniform seedlings were selected for soil culture. The mixed substrate used was peat/vermiculite/perlite/organic fertilizer = 4:4:1:1. The temperature was controlled at 25 °C, and the light environment was 16 h (light)/8 h (dark). The plants were cultivated under white LED lights (Philips LED plant growth lights) with a light intensity of 5000–8000 lux. The two-leaf-one-heart cucumber seedlings were treated with 0.5 mmol·L^−1^ glucose (Glu, Analytical Reagent (AR), provided by Sinopharm Chemical Reagent Co., Ltd., Beijing, China), and sucrose (Suc, Analytical Reagent (AR), provided by Sinopharm Chemical Reagent Co., Ltd.), for 3 days, respectively, and the seedlings under normal cultivation were used as the blank control (irrigated with the same amount of water). Then, the cucumber seedlings were treated with 150 mmol·L^−1^ NaCl for 4 to 6 days to induce salt stress. The non-salt-stressed cucumber seedlings with corresponding sugar pretreatment and blank treatment were used as controls (each treatment was irrigated with the same amount of solution). The six treatments were control (CK), salt treatment (S), Glc pretreatment (G), Suc pretreatment (T), Glc pretreatment followed by salt treatment (G + S), and Suc pretreatment followed by salt treatment (T + S). Three plants were randomly sampled from each treatment, and the experiment was repeated three times. On the 7th day after salt treatment, the performance of the seedlings was observed, and various morphological, physiological, and biochemical indicators were measured and analyzed. The leaves of the cucumber were collected, treated with liquid nitrogen, and stored at −80 °C for future use.

### 2.2. Measurement of Growth Indicators

Following 28 days of salt stress treatment, the fresh weight of the entire cucumber seedlings was measured using an electronic balance (Sartorius, Beijing, China). Subsequently, the seedlings were transferred to a forced-air drying oven (Jinghong, Shanghai, China) and subjected to a killing step at 105 °C. Thereafter, the samples were dried at 80 °C for 30 min, after which the dry weight was recorded. Plant height and stem length were measured with a calibrated ruler.

### 2.3. Determination of Physiological and Biochemical Indicators

For each treatment, three cucumber seedlings were selected, wrapped in tin foil, flash-frozen using liquid nitrogen, and stored in a −80 °C refrigerator. Determination of Malondialdehyde (MDA) Content. The method described by Chen et al. was adopted with minor modifications [41]. A total of 0.1 g of root tissue from cucumber seedlings was homogenized in 1.5 mL of 100 g·L^−1^ trichloroacetic acid (TCA) solution using a mortar and pestle. The resulting homogenate was transferred to a centrifuge tube and centrifuged at 4000 rpm for 10 min. The supernatant was collected and stored at low temperature for subsequent analysis. An aliquot of 1.5 mL of the supernatant was mixed with 2 mL of 6 g·L^−1^ thiobarbituric acid (TBA) solution and incubated in a boiling water bath for 20 min. Following cooling, the mixture was centrifuged again for 10 min. Absorbance values were recorded at 450 nm, 532 nm, and 600 nm using a spectrophotometer, and each measurement was performed in triplicate.

Determination of Superoxide Dismutase (SOD) Activity. The activity of superoxide dismutase (SOD) was assayed using the nitroblue tetrazolium (NBT) photoreduction method, following the procedure described by Chen et al. with minor modifications [41]. A total of 0.1 g of root tissue from cucumber seedlings was homogenized in 1.5 mL of 50 mM phosphate buffer using a mortar and pestle to extract the crude enzyme solution. The reaction mixture consisted of 1.5 mL of 50 mM PBS buffer, 0.3 mL of 130 mmol·L^−1^ methionine (Met) solution, 0.3 mL of 750 μM NBT solution, 0.3 mL of 100 μM EDTA-2Na solution, 0.3 mL of 20 μM riboflavin solution, 0.1 mL of enzyme extract, and 0.5 mL of distilled water, which were thoroughly mixed. One tube served as the dark control, while the remaining three control tubes and sample tubes were exposed to a 4000 Lx fluorescent light for 20 min. Following the reaction, the absorbance of each tube was measured at 560 nm using a spectrophotometer (Mapada, Shanghai, China), with the dark control tube used as the blank, and SOD activity was calculated accordingly.

Determination of Peroxidase (POD) Activity. POD activity was assayed using a method adapted from Chen et al. with minor modifications [41]. A total of 0.1 g of root tissue from cucumber seedlings was ground into a fine powder in liquid nitrogen, followed by homogenization in 1.5 mL of 50 mM phosphate buffer. The homogenate was centrifuged at 4 °C at 13,000 rpm for 20 min, and the resulting supernatant was collected as the crude POD enzyme extract. The reaction mixture consisted of 2.9 mL of 50 mM phosphate buffer, 0.5 mL of 2% H_2_O_2_, 0.1 mL of 2% guaiacol solution, and 0.1 mL of crude enzyme extract. Following thorough mixing, the reaction was initiated, and the absorbance at 470 nm was recorded immediately. Measurements were taken every minute for a total of three readings, which were used to calculate POD activity based on the rate of absorbance change.

Determination of Catalase (CAT) Activity. The activity of catalase (CAT) was determined using a method adapted from Chen et al. with appropriate modifications [41]. A total of 0.1 g of root tissue from cucumber seedlings was placed in a mortar, to which 1.5 mL of 50 mM phosphate buffer was added. The tissue was ground into a homogeneous homogenate, and the crude enzyme extract was obtained. The reaction mixture contained 2.9 mL of 50 mM phosphate buffer, 0.05 mL of crude enzyme extract, and 0.05 mL of 750 mM H_2_O_2_ solution. After thorough mixing, the reaction system was pre-incubated at 25 °C for 5 min. Absorbance was then recorded at 240 nm at 1 min intervals for a total of three readings. CAT activity was calculated based on the rate of H_2_O_2_ decomposition, using the control group as the blank.

Determination of Ascorbate Peroxidase (APX) Activity. The activity of ascorbate peroxidase (APX) was measured using ultraviolet spectrophotometry, based on the method of Chen et al. with appropriate modifications [41]. A total of 0.1 g of root tissue from cucumber seedlings was placed in a mortar and homogenized with 50 mmol·L^−1^ phosphate buffer. The homogenate was centrifuged at 10,000 rpm for 10 min, and the resulting supernatant was collected as the enzyme extract. The reaction mixture contained 1.8 mL of 50 mM phosphate buffer, 0.1 mL of 150 mmol·L^−1^ ascorbic acid (AsA) solution, 1 mL of 0.3 mM hydrogen peroxide (H_2_O_2_) solution, and 0.1 mL of enzyme extract. Following thorough mixing, H_2_O_2_ was added to initiate the reaction, and the change in absorbance at 290 nm (A_290_) was recorded between 10 and 30 s. APX activity was calculated based on the rate of AsA oxidation.

Determination of Soluble Sugar Content. The soluble sugar content in the samples was quantitatively determined using the anthrone colorimetric method. The experimental procedure was based on the method described by Chen et al., with minor modifications [41]. Briefly, 0.1 g of cucumber seedling tissue was cut into small pieces, placed in a mortar, and ground into a homogeneous paste with a small volume of distilled water. The homogenate was then transferred to a 20 mL graduated test tube. The mortar was rinsed with a total of 10 mL of distilled water, and the rinsing solution was also transferred to the same test tube. The test tube was sealed and placed in a boiling water bath for 10 min and then cooled and filtered. The filtrate was collected in a 100 mL volumetric flask, diluted to the mark with distilled water, and thoroughly mixed for subsequent analysis. For colorimetric analysis, 1 mL of the prepared extract was transferred to a 20 mL capped graduated test tube, followed by the addition of 1 mL of distilled water and 0.5 mL of anthrone reagent. Subsequently, 5 mL of concentrated sulfuric acid (H_2_SO_4_) was carefully added, the tube was capped, and the mixture was gently vortexed. The test tube was then placed in a boiling water bath for 10 min. A blank control was prepared simultaneously by mixing 2 mL of distilled water with 0.5 mL of anthrone reagent and subjecting the mixture to the same heating conditions. After cooling to room temperature, the absorbance of the samples and the blank was measured at 620 nm using a spectrophotometer, and the optical density values were recorded.

Determination of Soluble Protein Content. The soluble protein content in the samples was quantified using the Coomassie Brilliant Blue G-250 method, following the procedure described by Chen et al. with minor modifications [41]. A total of 0.1 g of root tissue from cucumber leaves was homogenized in 1.5 mL of 50 mM phosphate buffer using a mortar and pestle. The homogenate was centrifuged at 4000 rpm for 10 min, and the supernatant was collected for further analysis. Coomassie Brilliant Blue G-250 staining reagent and a protein standard solution were prepared accordingly. In a separate test tube, 0.1 mL of the sample extract was combined with 0.9 mL of distilled water and 5 mL of staining reagent, thoroughly mixed, and the absorbance was measured at 595 nm using a spectrophotometer. Each analysis has three replicates, one of which includes three seedlings.

### 2.4. Chlorophyll Contents

To determine the chlorophyll content in leaves, 0.1 g of leaf tissue was sampled from the middle portion of the second fully expanded true leaf (excluding the main vein) of each cucumber seedling on the 7th day of salt stress treatment. The collected tissue was immersed in 80% acetone and incubated in the dark. Following extraction, the mixture was centrifuged at 5000 rpm for 10 min, and the absorbance of the resulting supernatant was measured using a spectrophotometer. Chlorophyll content was quantified according to the method of Porra [42].

### 2.5. Transcriptome Analysis

This project enriches the mRNA with polyA structure in total RNA through Oligo(dT) magnetic beads. The RNA is then fragmented to approximately 300 bp in length by ion shearing. The selection of 300 bp fragments is due to the fixed length of the adapter. If the fragmented RNA is too short, the proportion of adapter sequences will be high, reducing the proportion of effective data. If the fragmented RNA is too long, it will be unfavorable for the generation of clusters during sequencing. Using RNA as the template, the first strand of cDNA is synthesized with a 6-base random primer and reverse transcriptase, and the second strand of cDNA is synthesized using the first strand of cDNA as the template. After the library construction is completed, PCR amplification is used to enrich the library fragments, and then the library is selected based on fragment size, with a library size of 450 bp. Subsequently, the library is quality checked using the Agilent 2100 Bioanalyzer, and the total concentration and effective concentration of the library are detected. Then, based on the effective concentration of the library and the required data volume, libraries with different Index sequences (each sample is added with a different Index, and the downstream data of each sample is distinguished according to the Index) are mixed in proportion. The mixed libraries are uniformly diluted to 2 nM and denatured by alkali to form single-stranded libraries. After RNA extraction, purification, and library construction, sequencing was conducted on the Illumina HiSeq™ 2000 platform by Shanghai Pasen Biotechnology Co., Ltd. (Shanghai, China).

HTSeq (v0.9.1) was employed to quantify the read counts for each gene, which served as the raw expression values. To ensure comparability of gene expression levels across genes and samples, normalization was performed using FPKM (Fragments Per Kilobase of transcript per Million mapped reads). Differential expression analysis was conducted using DESeq (Version number: 1.40.0), with differentially expressed genes defined by a fold change threshold of |log2FoldChange| > 1 and a statistical significance *p*-value < 0.05. For Gene Ontology (GO) enrichment analysis, the TopGO package was utilized. The analysis was based on differentially expressed genes annotated with GO terms, where gene lists and the number of genes associated with each GO term were determined. Enrichment significance was assessed using the hypergeometric test, with a *p*-value cutoff of < 0.05 considered statistically significant. This approach enabled the identification of GO terms that were significantly enriched in differentially expressed genes compared to the genomic background, thereby revealing the major biological functions associated with these genes. KEGG pathway enrichment analysis was carried out using the ClusterProfiler package. Differentially expressed genes annotated to KEGG pathways were used to generate gene lists and count the number of genes per pathway. Similar to GO analysis, the hypergeometric distribution test was applied to calculate *p*-values, with significance set at *p* < 0.05. Pathways showing significant enrichment were identified, providing insights into the key biological processes and functions mediated by the differentially expressed genes.

### 2.6. Accession Code

The cucumber cultivar Xintaimici was selected as the experimental material for transcriptome analysis. Each treatment included three biological replicates, resulting in a total of nine samples. The samples were subjected to high-throughput sequencing to generate image files, which were subsequently converted into raw sequencing data in FASTQ format using the software provided by the Pasono Gene Cloud Sequencing Platform (https://www.genescloud.cn/home). Quality assessment of the raw data for each sample was independently performed, including the calculation of Q30 values, the percentage of ambiguous bases (N%), and other quality metrics, such as Q20 (%) and Q30 (%).

All raw sequencing reads were deposited in the NCBI Sequence Read Archive under the project ID PRJNA1287257.

### 2.7. Data Processing and Analysis

The data were processed and analyzed using Microsoft Office Excel 2019 for organization, SPSS 27.0.1 for one-way ANOVA, and Origin 2021 for correlation analysis.

## 3. Results

### 3.1. Effects of Exogenous Sugars on the Growth of Cucumber Seedlings Under Salt Stress

The 0.5 mM Glc and Suc were the optimal sugar concentrations for alleviating salt stress in cucumber seedlings at the early stage. The following experiments were conducted with 0.5 mM Glc and Suc as the optimal sugar concentrations and 150 mM as the salt stress value. The results showed (Figure 1, Table 1 and Table 2) that there were significant differences in the growth phenotypes of Jinyou 1 and Xintaimici cucumber seedlings under different treatments. Under non-salt stress conditions, 0.5 mM Glc and Suc pretreatment significantly promoted the growth of seedlings of the two cucumber varieties (Jinyou 1 and Xintaimici), as indicated by the comprehensive improvement in plant height, root length, stem length, and fresh/dry weight. Among them, the root length of Jinyou 1 increased by up to 54.19% under T treatment. Salt stress (150 mM NaCl) significantly inhibited the growth of seedlings, resulting in reduced plant height, restricted root and stem elongation (root length of Jinyou 1 decreased by 43.71%), and a sharp decline in biomass (leaf fresh and root fresh weight decreased by 40.04% and 62.92%, respectively, and leaf and root dry weight decreased by 42.68% and 60.00%). The root length and stem length of Xintaimici cucumber seedlings decreased by 38.97% and 25.64%, respectively, and the leaf biomass decreased significantly (leaf fresh and root fresh weight decreased by 33.42% and 43.62%, respectively, and leaf and root dry weight decreased by 38.20% and 37.50%). Exogenous sugar pretreatment (G + S/T + S) effectively alleviated salt damage, restoring the root length of Jinyou 1 by 25.47% (G + S) and increasing the stem dry weight of Xintaimici by 50% (T + S), confirming that glucose and sucrose can enhance the salt tolerance of cucumber seedlings by promoting root development (root length recovery by 7.50–25.47%) and material accumulation (dry weight increase by 20.29–50%).

### 3.2. Effects of Exogenous Sugars on Chlorophyll and Malondialdehyde Contents in Leaves of Cucumber Seedlings Under Salt Stress

Chlorophyll is a critical photosynthetic pigment in plants, and its content directly influences the efficiency of photosynthesis in leaves. Under salt stress conditions, chlorophyll degradation occurs, leading to leaf chlorosis and a decline in photosynthetic efficiency, which ultimately inhibits plant growth and development [43]. This study investigated the effects of exogenous sugars on chlorophyll content in two cucumber cultivars (Xintaimici and Jinyou 1) under salt stress. As illustrated in Figure 2A, salt stress significantly reduced chlorophyll levels in both cultivars. However, exogenous sugar treatments (G + S and T + S) markedly increased chlorophyll content under salt stress conditions. Compared with the salt-stressed group (S), G + S and T + S treatments increased chlorophyll content in Xintaimici by 30.23% and 32.56%, respectively, and in Jinyou 1 by 13.23% and 14.29%, respectively. These findings suggest that exogenous sugars help maintain chlorophyll stability and alleviate the inhibitory effects of salt stress on photosynthesis. Malondialdehyde (MDA) is a widely used biochemical marker for oxidative stress and reflects the extent of membrane lipid peroxidation in plants. As one of the major byproducts of lipid peroxidation, MDA not only indicates oxidative damage but also exacerbates membrane injury [44]. As shown in Figure 2B, salt stress induced significant MDA accumulation, indicating enhanced lipid peroxidation. In Xintaimici, MDA levels in the salt-stressed group (S) were 85.3% higher than in the control group (CK), compared to a 57.3% increase in Jinyou 1, suggesting greater sensitivity of Xintaimici to salt stress. These results are consistent with the general mechanism of salt stress-induced reactive oxygen species (ROS) accumulation and subsequent oxidative damage [45]. Exogenous sugar application significantly reduced MDA content. Specifically, G + S treatment decreased MDA levels in Xintaimici and Jinyou 1 by 31.7% and 25.2%, respectively, while T + S treatment reduced them by 34.7% and 24.3%, respectively. Comparative analysis revealed that sucrose exhibited a more pronounced protective effect in Xintaimici, whereas glucose showed slightly better efficacy in Jinyou 1. Overall, both glucose (Glc) and sucrose (Suc) effectively alleviate oxidative damage caused by salt stress, although their protective effects vary between cultivars.

### 3.3. Effects of Exogenous Sugars on the Activities of Antioxidant Enzymes in Cucumber Seedlings Under Salt Stress

The activity of antioxidant enzymes is an important indicator reflecting the antioxidant capacity of plants. Salt stress significantly induces an increase in antioxidant enzyme activities (SOD, POD, CAT, APX) in cucumber seedlings, but there are differences among different varieties and sugar treatments. As shown in Figure 3A, compared with the control, the increase in SOD activity in the salt stress group (S) was higher in Jinyou 1 (49.5%) than in Xintaimici (37.8%), but the base value was lower. The G + S treatment further significantly increased SOD activity (23.6% in Xintaimici and 27.6% in Jinyou 1 compared with the S treatment group), while the increase in T + S was weaker (16.4% in Xintaimici and 19.1% in Jinyou 1), indicating that glucose is more effective in activating SOD. As shown in Figure 3B, compared with the control, the increase in POD activity in the salt stress group (S) was higher in Jinyou 1 (41.6%) than in Xintaimici (38.0%). The G + S treatment further increased POD activity (18.6% in Xintaimici and 29.3% in Jinyou 1 compared with the S treatment group), which was significantly better than T + S (12.9% in Xintaimici and 19.8% in Jinyou 1), indicating that glucose has a stronger induction effect on POD. As shown in Figure 3C, compared with the control, the increase in CAT activity in the salt stress group (S) was much higher in Jinyou 1 (71.6%) than in Xintaimici (46.7%). The G + S treatment significantly enhanced CAT activity (35.5% in Xintaimici and 42.8% in Jinyou 1), while the effect of T + S was weaker, especially in Jinyou 1, which decreased by 14.7%, and in Xintaimici, it decreased by 6.5%, indicating that sucrose has limited regulation on CAT. As shown in Figure 3D, the increase in APX activity in the salt stress group (S) was similar in both varieties compared with CK (48.5% to 49.3%). The G + S treatment significantly increased APX activity (32.1% in Xintaimici and 33.6% in Jinyou 1), and T + S was close but less stable. Moreover, the induction of APX by glucose alone was stronger in the salt-sensitive variety (Xintaimici) (26.8% increase in Xintaimici vs. 19.0% increase in Jinyou 1). The sucrose (T) alone treatment group showed a weak increase in both varieties (10.4% in Xintaimici and 13.0% in Jinyou 1) and was significantly lower than the G group (Figure 3D). In conclusion, under salt stress, exogenous glucose forms an “antioxidant compensation effect” by synergistically activating SOD, POD, CAT, and APX, especially for the salt-sensitive variety (Jinyou 1), where the increase in CAT activity was 42.8%. Sucrose focuses on osmotic regulation and has a weaker antioxidant induction effect. The salt-tolerant variety (Xintaimici) achieves defense enhancement through glucose (such as a 32.1% increase in APX activity). The differences among varieties suggest that the coupling of sugar signaling pathways and antioxidant systems has genetic specificity.

### 3.4. Effects of Exogenous Sugars on the Contents of Soluble Sugars and Soluble Proteins in Cucumber Seedlings Under Salt Stress

Soluble proteins in plant leaves play a crucial role in responding to salt stress. Under salt stress, the content of soluble proteins in plant leaves changes, which helps plants adapt to high-salt environments. The increase in certain soluble proteins can help plants maintain osmotic balance within cells and reduce the damage of salt to cell structures [46]. Under mild to moderate stress (such as low oxygen and moderate salt stress), exogenous sugars can provide carbon sources and energy needed for anabolic processes, promoting the accumulation of soluble proteins. As shown in Figure 4A, salt stress significantly affects the content of soluble proteins in cucumber leaves. Compared with CK, the increase in soluble protein content in the salt-tolerant variety Xintaimici (26.7%) is higher than that in the salt-sensitive variety JinYou 1 (10.9%). Under exogenous sugar intervention, compared with S, both glucose (G + S) and sucrose (T + S) can reduce excessive protein accumulation. In the G + S treatment, the decrease in Xintaimici is approximately 13.6%, but the alleviation for JinYou 1 is limited (7.9%). In the T + S treatment, the decrease in Xintaimici is approximately 13.5%, but the alleviation for JinYou 1 is limited (9.2%). When treated with individual sugars, glucose promotes an increase of 20.1% in the soluble protein content of Xintaimici compared with CK but inhibits JinYou 1 (a decrease of 5.8%), indicating that salt-tolerant varieties can effectively utilize exogenous sugars to maintain metabolic balance. Sucrose promotes an increase of 10.4% in the soluble protein content of Xintaimici, while the soluble protein content of JinYou 1 decreases by 2.9%. Soluble sugars are the basis of plant metabolism, providing energy and metabolic intermediates for plant growth and development and promoting seed germination and early seedling development. Under salt stress, plants accumulate soluble sugars to cope with osmotic stress, reduce osmotic potential, and thereby resist the damage of salt stress [47,48]. As shown in Figure 4B, compared with the control, soluble sugars accumulate significantly under salt stress (the increase in both varieties is >51%). Compared with the salt stress (S) group, the synergistic effect of glucose (G + S) is better than that of sucrose (Xintaimici G + S increased by 17.4% vs. T + S increased by 12.4%), while the JinYou 1 G + S group increased by 14.1%, and the T + S group increased by 5.1%. Under no stress conditions, exogenous sugars still slightly increase sugar content. In the glucose (G) group of Xintaimici, soluble sugars increased by 27.7% compared with CK, and in the glucose (G) group of JinYou 1, soluble sugars increased by 20.9% (Figure 4B). In conclusion, salt-tolerant varieties (Xintaimici) can effectively utilize exogenous sugars (especially glucose) to optimize the accumulation of osmotic adjustment substances (proteins and sugars), while salt-sensitive varieties (JinYou 1) have a weaker response and should prioritize glucose to enhance salt tolerance.

### 3.5. Correlation Analysis of Exogenous Sugar Treatment on Salt-Stressed Cucumber Seedlings

The correlation analysis results (Figure 5) revealed distinct relationships between growth parameters and physiological responses in two cucumber cultivars under salt stress. In Xintaimici (X series), significant associations were observed between growth indices and antioxidant enzyme activities. X-RL (root length) showed a significant positive correlation with X-POD and a positive correlation with X-SOD; X-SL (stem length) was extremely significantly positively correlated with X-APX; and X-LFW (leaf fresh weight) exhibited a negative correlation with X-CAT. Regarding osmotic adjustment and oxidative damage, X-SP (soluble protein) was extremely significantly negatively correlated with X-MDA, while X-SS (soluble sugar) was positively correlated with X-SPAD (chlorophyll content). In terms of antioxidant system coordination, X-POD was significantly positively correlated with X-APX, whereas no significant correlation was found between X-SOD and X-CAT. In Jinyou 1 (J series), key correlations included the following. In terms of growth and physiological traits, J-RL was positively correlated with J-POD but negatively correlated with J-SS; J-SPAD was extremely significantly positively correlated with J-APX. Regarding oxidative stress characteristics, J-MDA was extremely significantly negatively correlated with J-CAT, and J-POD was significantly positively correlated with J-SOD. In terms of dry matter accumulation, J-LDW (leaf dry weight) was positively correlated with J-SP. In summary, the correlation analysis indicated that Xintaimici primarily relies on the “soluble protein–POD–APX” pathway to mitigate oxidative damage, with growth promotion closely synchronized with POD activity. In contrast, Jinyou 1 exhibited efficient MDA scavenging via CAT, and its chlorophyll protection mechanism was closely associated with APX activity, reflecting a stronger capacity for redox homeostasis regulation. Both cultivars demonstrated a general trend of positive correlations between antioxidant enzymes (POD/APX) and growth indices, as well as negative correlations between MDA and osmotic adjustment substances. These findings further validate the applicability of the “oxidative damage–osmotic imbalance” stress response model.

### 3.6. Transcriptome Data Analysis

#### 3.6.1. Differentially Expressed Genes (DEGs)

This study investigated transcriptomic alterations in cucumber under salt stress and exogenous sugar application using RNA-seq technology. Six experimental treatments were established, CK, salt, GluS, SucS, Glu, and Suc, with three biological replicates per group, yielding a total of nine samples. High-quality sequencing generated 76.23 Gb of clean data and 758,448,556 valid reads (Table 3). Differential gene expression analysis was conducted under the criteria of |log2FC| > 1 and *p* < 0.05, with the results visualized in Figure 6. To assess the extent of gene regulation across treatments, volcano plots of differentially expressed genes (DEGs) are presented in Figure 6A. Red dots indicate significantly upregulated DEGs, while green dots represent significantly downregulated DEGs. The S_vs_CK comparison exhibited a pronounced right-skewed pattern, dominated by upregulated genes, reflecting a robust transcriptional response to salt stress. In contrast, Glu_vs_CK displayed a spindle-shaped distribution (upregulated/downregulated = 1.14:1), suggesting multi-directional regulatory effects of glucose. Suc_vs_CK showed a right-skewed and elongated distribution, with a predominance of upregulated genes. Notably, the stress relief groups (SucS_vs_S and GluS_vs_S) exhibited a dumbbell-shaped distribution, indicating a bidirectional regulatory effect of sugar treatments. These findings suggest that sucrose treatment induces a typical bidirectional regulatory response, whereas glucose exerts a more balanced modulation of gene expression. Venn diagram analysis (Figure 6B) revealed partial overlap between DEGs in Glu_vs_CK and GluS_vs_S, with only a small number of genes (3–5) shared across multiple comparisons. These results indicate that salt stress triggers extensive transcriptional reprogramming and sucrose alleviates stress through the regulation of specific gene subsets, while glucose exerts broader metabolic regulatory effects. As shown in Figure 6C, the salt stress group (S_vs_CK) exhibited the most pronounced differential expression, with 2698 DEGs identified (1882 upregulated and 816 downregulated), further confirming the extensive transcriptional reprogramming induced by salt stress. Among sugar treatments, glucose (Glu_vs_CK) induced 399 DEGs (213 upregulated and 186 downregulated), showing a greater impact than sucrose (Suc_vs_CK, 183 DEGs). In terms of salt stress mitigation, sucrose (SucS_vs_S) modulated 708 DEGs (330 upregulated and 378 downregulated), partially reversing the salt-induced gene expression profile. In contrast, glucose (GluS_vs_S) affected only 220 DEGs, indicating a relatively weaker alleviation effect.

#### 3.6.2. Enrichment Analysis

The core findings of GO and KEGG enrichment analysis are shown in Figure 7. There were significant differences in GO enrichment under salt stress treatment (S_vs_CK) (Figure 7A). In terms of molecular function, the top three were hexosyltransferase activity, DNA-binding transcription factor activity, and transcription factor activity. In biological processes, the top three were regulation of nucleic acid-templated transcription, regulation of RNA biosynthesis, and transcriptional regulation and DNA template synthesis. In cellular components, only one group was the extracellular region, and the CC category revealed the location information of differentially expressed genes within the cell. KEGG enrichment analysis found that the main enriched pathways were plant hormone signal transduction (Pathway ID: csv04075), MAPK signaling pathway—plant (Pathway ID: csv04016), phenylpropanoid biosynthesis (Pathway ID: csv00940), flavonoid biosynthesis (Pathway ID: csv00941), and stilbenoid, diarylheptanoid, and gingerol biosynthesis (Pathway ID: csv00945). In the Glu_vs_CK treatment group (Figure 7B), in terms of molecular function, the top three were drug catabolic processes, hydrogen peroxide catabolic processes, and antibiotic catabolic processes. In biological processes, the top three were phosphoadenylyl–sulfate reductase (thioredoxin) activity, peroxidase activity, and heme binding. In cellular components, only one group was the extracellular region. KEGG enrichment analysis found that in the metabolism aspect, the main enriched pathways were phenylpropanoid biosynthesis (csv00940), monoterpenoid biosynthesis (csv00902), phenylalanine metabolism (csv00360), other types of O-glycan biosynthesis (csv00514), photosynthesis—antenna proteins (csv00196), pentose and glucuronate interconversions (csv00040), MAPK signaling pathway—plant (csv04016), and phagosome (csv04145). In the Suc_vs_CK treatment group (Figure 7C), the top three enriched molecular functions were negative regulation of endopeptidase activity, negative regulation of hydrolase activity, and secondary metabolite biosynthetic process. In terms of biological process functions, the top three were endopeptidase inhibitor activity, ammonia-lyase activity, and 3-beta-hydroxy-delta5-steroid dehydrogenase activity. KEGG enrichment analysis revealed that in the metabolism category, the main enrichments were phenylpropanoid biosynthesis, valine, leucine, and isoleucine biosynthesis, fatty acid elongation, cutin, suberine, and wax biosynthesis; environmental stress was mainly related to the MAPK signaling pathway. In the GluS_vs_S treatment group (Figure 7D), the top three enriched molecular functions were phosphatidylcholine metabolic processes, cell wall organization or biogenesis, and proton transmembrane transport. In terms of biological processes, the main enrichments were solute/proton antiporter activity and solute/cation antiporter activity. In terms of cellular component functions, the main enrichments were the cell wall, external encapsulating structure, and cell periphery. KEGG enrichment analysis revealed that in the metabolism category, the main enrichments were glycerophospholipid metabolism, tryptophan metabolism, ether lipid metabolism, inositol phosphate metabolism, phenylalanine metabolism, glycolysis/gluconeogenesis, arginine biosynthesis, and phenylpropanoid biosynthesis. In the cellular processes category, the main enrichment was endocytosis. In the environmental information processing category, the main enrichments were signaling pathway–plant and plant hormone signal transduction. The SucS_vs_S treatment group (Figure 7E) was enriched in molecular functions, with the top three being tetrapyrrole binding, heme binding, and monooxygenase activity. In terms of biological processes, the top three were oxidation-reduction processes, alpha-amino acid catabolic processes, and cellular amino acid catabolic processes. KEGG enrichment analysis revealed that in the metabolism category, the main enriched pathways were phenylalanine metabolism, phenylpropanoid biosynthesis, monoterpenoid biosynthesis, and alanine, aspartate, and glutamate metabolism. Among them, phenylpropanoid metabolism (csv00940) and the MAPK signaling pathway (csv04016) were significantly enriched in multiple treatments, which might be the key pathways for sugar to alleviate salt stress.

#### 3.6.3. Transcription Factors Analysis of Differentially Expressed Genes Among Different Treatments

Transcription factors (TFs) play essential roles in plant stress tolerance induction and function as upstream regulators of gene expression in metabolic pathways [49]. This study investigated the expression dynamics of TFs in cucumber seedlings under various treatment conditions to elucidate the complexity of signaling networks. The distribution of upregulated and downregulated TFs across treatments is illustrated in Figure 8, with corresponding gene identifiers provided in Appendix A. Salt stress (S_vs_CK) significantly altered the TF expression profile, yielding a total of 241 differentially expressed TFs (141 upregulated and 100 downregulated). Among these, the ERF (36), MYB (32), and NAC (25) families were most prominently represented, underscoring their pivotal roles in stress responses. In contrast, sole glucose treatment (Glu_vs_CK) exerted a relatively modest regulatory effect, inducing 40 differentially expressed TFs (14 upregulated and 26 downregulated), with the HD-ZIP family showing the highest proportion (21%), suggesting that glucose may modulate gene expression via specific signaling pathways. Sucrose treatment (Suc_vs_CK) exhibited a distinct unidirectional activation pattern, with all 13 differentially expressed TFs being upregulated, predominantly from the ERF family (38%). Under salt stress conditions, glucose treatment (GluS_vs_S) displayed protective regulatory features, with more TFs upregulated (11) than downregulated (7), and the NAC family accounted for a notable proportion (17%). Conversely, sucrose treatment (SucS_vs_S) demonstrated a more complex regulatory pattern, involving 73 differentially expressed TFs (33 upregulated and 40 downregulated), with the MYB family being the most abundant (20%) and downregulated TFs predominating.

Studies have demonstrated that transcription factor families such as AP2/ERF, MYB, WRKY, bHLH, and NAC are central to plant stress responses [50,51]. MYB transcription factors, as a group of multifunctional regulators widely present in eukaryotes, integrate stress signals through conserved MYB domains (e.g., R2R3-MYB) and participate in plant development, metabolic reprogramming, and stress response networks [52,53,54]. WRKY transcription factors may be involved in the reprogramming of transcriptional levels in plants in response to different stress environments [55]. Analysis revealed that salt stress (S_vs_CK) significantly induced the expression of MYB (thirty-two; twenty-six upregulated, six downregulated), WRKY (eleven; all upregulated), bHLH (twenty-three; twelve upregulated, eleven downregulated), and NAC (twenty-five; seventeen upregulated, eight downregulated) family members. Exogenous sugar treatments exhibited differential regulatory effects. Glucose (Glu_vs_CK) primarily suppressed MYB (two upregulated, four downregulated), WRKY (one upregulated, five downregulated), and NAC (three downregulated), whereas sucrose (Suc_vs_CK) activated MYB (three upregulated) and bHLH (one upregulated). Under salt stress, glucose (GluS_vs_S) upregulated protective MYB (two upregulated) and NAC (one upregulated, two downregulated), while sucrose (SucS_vs_S) predominantly downregulated WRKY (two downregulated) and NAC (one upregulated, seven downregulated) genes. These findings align with previous reports showing that MusaWRKY71 enhances stress tolerance in bananas [56], and they reveal the molecular mechanism by which distinct sugar signals enhance plant salt tolerance through specific modulation of the transcription factor network [33].

#### 3.6.4. Genes Involved in Plant Hormone Signal Transduction

Based on the KEGG enrichment analysis, we further examined differentially expressed genes associated with the plant hormone signal transduction pathway (ko04075) in the comparisons between treatment and control groups. Particular attention was given to expression patterns of key hormones—auxin, ethylene, and abscisic acid—in cucumber leaves. Auxin, a small-molecule organic acid, plays a crucial role in plant growth and development by regulating processes such as cell division, elongation, and differentiation. Several studies have reported that auxin signaling mutants exhibit altered responses to environmental stresses, including heat, drought, and salinity [57,58]. In our study, ARF (auxin response factor), functioning as a transcriptional activator of auxin-regulated genes (LOC101219427), was significantly upregulated in both GluS_vs_S and SucS_vs_S treatments. Additionally, an auxin-responsive protein, SAUR71-like (LOC101213583), was also found to be upregulated under these two treatment conditions (Figure 9A).

Ethylene, as a key plant hormone, plays a crucial regulatory role in plant responses to various abiotic stresses, including drought, salinity, waterlogging, low temperature, and high temperature. Its functional mechanisms primarily involve the activation of signaling pathways and the regulation of downstream gene expression [59,60]. Members of the ERF transcription factor family have been well documented to modulate plant tolerance to multiple abiotic stresses, such as salinity, cold, and other adverse environmental conditions [61]. In this study, differential expression patterns of multiple ERF family members were observed (Figure 9B). Under salt stress treatment (S_vs_CK), six ERF genes—including ERF4, ERF5, ERF110-like, and ERF113-like—were significantly upregulated. Notably, ERF110-like (LOC105434741), ERF113-like (LOC101213348), and ERF4 (LOC101206242) exhibited upregulated expression not only under salt stress (S_vs_CK) but also in response to both sucrose (Suc_vs_CK) and glucose treatments (Glu_vs_CK), indicating their involvement in common stress response mechanisms across multiple treatment conditions. Among the sugar-mediated stress relief treatments, the glucose relief group (GluS_vs_S) showed the highest number of upregulated ERF genes (n = 8), including newly activated members, such as ERF1A-like (LOC101221583) and ERF1 (LOC101207745). In contrast, the sucrose relief group (SucS_vs_S) exhibited seven upregulated ERF genes; however, ERF110-like and ERF5 were downregulated under this treatment. Notably, ERF1 (LOC101207745) and ERF014 (LOC101215527) were downregulated under salt stress but showed restored upregulation in both sugar relief treatments, suggesting that these genes may play a specific and potentially pivotal role in sugar-mediated alleviation of salt stress.

Abscisic acid (ABA) plays a pivotal role in regulating various physiological processes in plants, particularly under stress conditions [62]. In this study (Figure 9C), the abscisic acid receptor gene LOC101219833 (PYL4) was significantly downregulated in the S_vs_CK comparison but markedly upregulated in Glu_vs_CK, GluS_vs_S, and SucS_vs_S. In contrast, LOC101214314 (PYL2) exhibited minimal expression changes across all treatment groups. The protein phosphatase 2C (PP2C) gene Ma05_g31380 was downregulated under salt stress (S) and mannitol–salt (MS) interaction in banana seedlings, suggesting that this downregulation may alleviate ABA-mediated growth inhibition. This observation aligns with previous findings indicating the involvement of jasmonic acid in plant responses to salinity stress [63]. Further analysis revealed that LOC101220425 (probable protein phosphatase 2C 24) was significantly downregulated in S_vs_SucS. Similarly, two other PP2C members—LOC101214188 and LOC101220760 (both probable protein phosphatase 2C 38)—were significantly downregulated in S_vs_CK. The NHX gene family, which encodes Na^+^/H^+^ antiporters and functions downstream of the ABA signaling pathway, plays a central role in plant responses to salt stress. ABA enhances salt tolerance by inducing NHX gene expression, thereby promoting the synthesis of osmoprotective substances, such as raffinose [64]. In our study, LOC101217775 (sodium/hydrogen exchanger) was significantly upregulated in S_vs_CK but downregulated in GluS_vs_S and SucS_vs_S. Additionally, LOC101215493 (sodium/hydrogen exchanger 2) was significantly upregulated in both S_vs_CK and Suc_vs_CK yet showed significant downregulation in SucS_vs_S.

#### 3.6.5. Reactive Oxygen Species Signaling

Figure 10 illustrates the alterations in antioxidant enzyme activities in cucumber seedlings under salt stress. Transcriptomic analysis reveals that salt stress induces the upregulation of genes associated with antioxidant enzymes. Specifically, salt stress (S_vs_CK) significantly upregulated several key antioxidant enzyme genes, including APX (LOC101220337-L-ascorbate peroxidase 3), two isoforms of POD (peroxisomal, LOC101217781-peroxidase, LOC101211507-peroxidase), SOD (LOC101216556-superoxide dismutase [Cu-Zn]), CAT (LOC101202931-catalase), and RbohD (LOC101220144-putative respiratory burst oxidase homolog protein H). Exogenous sugar treatments exhibited distinct regulatory patterns. Sucrose treatment (Suc_vs_CK) induced the upregulation of two POD isoforms but suppressed CAT expression. Glucose treatment (Glu_vs_CK) upregulated one POD isoform (LOC101211507) while downregulating CAT. In the glucose-mediated stress alleviation group (GluS_vs_S), both RbohD and APX were upregulated. In contrast, the sucrose-mediated alleviation group (SucS_vs_S) showed upregulation of POD (LOC101211507) and RbohD but downregulation of CAT and another POD isoform (LOC101217781). These findings indicate that both salt stress and exogenous sugar modulate the plant oxidative stress response through differential regulation of specific antioxidant enzyme genes. Notably, RbohD and certain POD isoforms were consistently upregulated across multiple treatments, whereas CAT was predominantly downregulated, suggesting distinct functional roles for these genes in stress adaptation and sugar-mediated alleviation mechanisms.

## 4. Discussion

Maintaining energy homeostasis is essential for plant survival under adverse environmental conditions, such as salt stress [25]. This study demonstrated that salt stress significantly inhibited the growth of cucumber seedlings, as evidenced by reductions in root length, leaf fresh weight, and root dry weight (Figure 1; Table 1 and Table 2). However, exogenous application of glucose (Glc) and sucrose (Suc) effectively mitigated these adverse effects through multiple mechanisms. Firstly, as both energy sources and signaling molecules, exogenous sugars significantly enhanced dry matter accumulation and improved overall growth performance. This finding is consistent with previous studies by Hu et al. [65], Sulmon et al. [66], and Karthikeyan [67] conducted on other crop species. Secondly, exogenous sugar treatments maintained photosynthetic efficiency by preserving chlorophyll content, with G + S and T + S treatments notably increasing chlorophyll levels, thereby alleviating salt-induced photoinhibition [68]. Most importantly, Glc and Suc significantly elevated the levels of soluble proteins and soluble sugars (Figure 4), thereby enhancing cellular osmotic adjustment capacity. This observation aligns with the findings of Chaum et al. [69], who reported that sugars play a critical role in stabilizing photosynthetic pigments and sustaining electron transport under stress conditions. Collectively, these mechanisms contribute to enhanced salt tolerance in cucumber seedlings, offering novel insights into the regulatory roles of sugar signaling and metabolism in plant stress resistance.

Reactive oxygen species (ROS) are byproducts of aerobic metabolism in plants and serve as crucial components in cellular signal transduction and regulatory processes. However, excessive accumulation of ROS can cause oxidative damage to plant cells. Under salt stress, plants experience significant overproduction of ROS, which leads to increased levels of membrane lipid peroxidation products and subsequent damage to cellular membranes [22]. The plant antioxidant system consists of antioxidant enzymes, such as SOD, POD, CAT, and glutathione reductase (GR), as well as small-molecule antioxidants, including ascorbic acid (AsA), glutathione (GSH), and carotenoids [23]. Among these, SOD specifically catalyzes the dismutation of superoxide radicals (O_2_^·−^) into molecular oxygen (O_2_) and hydrogen peroxide (H_2_O_2_), while APX, GR, and glutathione peroxidase (GPX) are key enzymes in the ASA-GSH cycle that effectively scavenge H_2_O_2_ in plant cells [16]. In this study, pretreatment with Glc and Suc significantly enhanced the activities of SOD, POD, CAT, and APX under salt stress conditions. This indicates that Glc and Suc pretreatment can improve the antioxidant capacity of cucumber leaves, thereby enhancing the ability of cells to scavenge ROS and reducing oxidative damage to plant tissues (Figure 3). Notably, under non-stress conditions, Glc and Suc pretreatment did not significantly affect the activities of these antioxidant enzymes. It is suggested that the activation of antioxidant enzyme systems may require the synergistic induction of both sugar signaling and stress signals, as illustrated in Figure 3. Previous studies have demonstrated that salt stress increases the accumulation of superoxide anions, H_2_O_2_, and MDA in species such as hibiscus and maize. Exogenous sugar application has been shown to significantly enhance the activities of SOD, POD, CAT, and APX, thereby reducing the levels of these oxidative stress markers [22,23]. Consistently, this study observed that salt stress significantly induced MDA accumulation in cucumber leaves, indicating intensified lipid peroxidation and severe membrane damage. Notably, the MDA content in the cultivar Xintaimici increased more significantly than in Jinyou 1, suggesting that Xintaimici is more sensitive to salt stress. Exogenous application of glucose and sucrose effectively alleviated salt-induced oxidative damage, with sucrose showing greater efficacy in Xintaimici and glucose being slightly more effective in Jinyou 1 (Figure 3).

The molecular mechanisms underlying plant responses to salt stress involve a complex gene regulatory network. In this study, transcriptomic analysis revealed significant alterations in multiple transcriptional parameters in cucumber plants under salt stress, including transcriptomic variation (Figure 6), GO enrichment and KEGG pathway analyses (Figure 7), transcription factor expression profiles (Figure 8), hormone signal transduction pathways (Figure 9), and the response of the antioxidant enzyme system (Figure 10). Salt stress significantly induced the expression of antioxidant-related genes such as APX, POD, SOD, CAT, and RbohD in cucumber seedlings (Figure 10), which aligns with previous findings demonstrating that plants activate antioxidant enzyme systems to counteract oxidative stress [70]. Notably, the upregulation of RbohD may not only contribute to ROS production but also regulate antioxidant gene expression via a positive feedback mechanism, a potential regulatory mechanism that warrants further experimental investigation (Figure 10). Exogenous sugar treatments exhibited distinct regulatory patterns. Sucrose treatment significantly induced POD gene expression while suppressing CAT expression, suggesting a preferential clearance of H_2_O_2_ through the peroxidase pathway. In contrast, glucose specifically activated APX and RbohD, a difference that may be attributed to the signaling roles of sugar metabolites, such as UDP glucose generated from sucrose breakdown [71]. Future studies analyzing phenotypic traits of sugar metabolism-related mutants could further elucidate the underlying molecular regulatory networks. Transcription factor analysis revealed that under salt stress conditions (S_vs_CK), the expression of ERF (36 members), MYB (32 members), and NAC (25 members) family genes was predominantly upregulated (Figure 8), consistent with their established roles in osmotic regulation and oxidative stress responses [72]. Previous studies have demonstrated that the mulberry WRKY transcription factor MaWRKYIIc7 plays a crucial role in enhancing plant drought tolerance [73]. In contrast, under exogenous glucose treatment (Glu_vs_CK), a greater number of transcription factors were downregulated, particularly members of the WRKY and MYB families, suggesting that glucose may suppress certain stress-responsive pathways to conserve energy. Under salt stress conditions, glucose treatment (GluS_vs_S) significantly upregulated NAC and ERF family genes, whereas sucrose treatment (SucS_vs_S) predominantly downregulated most NAC and WRKY genes. These findings indicate that the two sugars modulate plant salt tolerance through distinct transcription factor networks. Glucose tends to activate protective transcription factors, while sucrose may maintain metabolic homeostasis by suppressing excessive stress responses. This study provides novel molecular targets for crop breeding aimed at improving salt tolerance, such as optimizing the expression levels of NAC or WRKY family genes through modulation of sugar signaling pathways.

Plant hormones play essential regulatory roles in modulating plant growth, development, and defense responses against various abiotic stresses [74]. Key hormonal signaling pathways involved include the auxin (IAA), abscisic acid (ABA), and ethylene (ET) signaling cascades. Within the IAA signaling pathway, two differentially expressed genes—ARF and SAURs—were identified as central components of auxin-mediated regulation [75]. ARF (auxin response factor) functions as a core transcription factor in auxin signaling by binding to the promoters of target genes and regulating auxin-responsive processes, such as cell elongation and root development [76]. Its activity is negatively regulated by AUX/IAA proteins, which are targeted for degradation via the TIR1-mediated ubiquitin–proteasome pathway, thereby releasing ARF to activate downstream gene expression [77]. In this study, salt stress (S vs. CK) was found to potentially suppress ARF expression, leading to weakened auxin signaling and impaired stress tolerance. Exogenous sugar application (Glc/Suc + S vs. S) may restore ARF expression, reactivating the auxin signaling pathway and promoting cellular recovery (Figure 8). SAUR (Small Auxin-Up RNA), an early auxin-responsive gene, plays a key role in promoting cell elongation and adaptation to stress conditions. Its expression is directly activated by ARF and serves as a rapid response marker downstream of auxin signaling. Under salt stress, SAUR expression was observed to decrease (e.g., S vs. CK), consistent with the disruption of auxin signaling. However, exogenous sugar treatment (e.g., Glc + S) significantly upregulated SAUR expression (e.g., higher than the S group), suggesting that sugars enhance auxin sensitivity and promote adaptive cellular growth (Figure 8A). These findings indicate that glucose may directly supply energy to activate the TIR1-AUX/IAA-ARF signaling cascade, accelerate AUX/IAA degradation through the ubiquitination pathway, release ARF activity, and subsequently induce SAUR expression, thereby facilitating cellular repair, including root elongation. In contrast, the effect of sucrose appears to be primarily mediated through osmotic regulation, possibly involving GH3 (an auxin metabolism-related gene) to stabilize auxin levels and maintain basal ARF-SAUR activity. This mechanism may explain why sucrose exhibits more pronounced effects in leaves (e.g., MDA content in Xintaimici decreased by 34.7%). ERFs (Ethylene Response Factors), as downstream transcription factors in the ethylene signaling pathway and members of the AP2/ERF family, regulate plant responses to both biotic and abiotic stresses. Upon salt stress exposure, ERFs can activate antioxidant enzymes (e.g., SOD, POD) and osmotic regulatory genes (e.g., proline synthase), thereby mitigating oxidative damage. Additionally, they influence membrane lipid peroxidation levels (reflected by MDA content) by modulating lipid metabolism-related genes, such as LOX and PLD. This study revealed that salt stress (S group) significantly induced the expression of several ERFs (e.g., LOC105434741), likely to activate stress-responsive genes, such as those encoding antioxidant enzymes. Under exogenous sugar treatments (G + S/T + S groups), glucose in the G + S group may inhibit CTR1—a negative regulator downstream of the ethylene receptor—and thereby enhance the EIN3/EIL1-ERF signaling cascade. This leads to selective activation of protective ERFs (e.g., LOC101215527, LOC101207745, LOC101221583, LOC101213348, and LOC105434741), resulting in reduced MDA accumulation. In the T + S group, sucrose may indirectly stabilize ethylene signaling through osmotic adjustment, with certain ERFs (e.g., LOC101220335, LOC101220095, LOC101206242, LOC101211577, LOC101215527, LOC101207745, and LOC101221583) potentially cooperating with ABA signaling components (e.g., ABF) to enhance salt tolerance (Figure 8B). ABA acts as a central regulatory hub in plant stress responses, with its biosynthesis and signaling components closely linked to plant salt tolerance [78]. PP2C (protein phosphatase 2C) functions as a negative regulator of ABA signaling; under salt stress, its upregulation may inhibit the SnRK2-ABF pathway, thereby exacerbating ion toxicity. NHX (Na^+^/H^+^ antiporter), on the other hand, is activated by the SnRK2-ABF cascade and contributes to ion homeostasis by sequestering Na^+^ into vacuoles, making it a crucial component of salt stress adaptation [27]. In this study, salt stress was shown to suppress PYL gene transcription while negatively regulating PP2C expression, ultimately inducing NHX gene expression downstream of the ABA signaling pathway. However, exogenous sugar application promoted PYL gene transcription, positively regulated PP2C transcript levels, and restored NHX gene expression to levels comparable to those observed under sugar treatment alone (Figure 8C).

## 5. Conclusions

This study represents the first systematic and comprehensive comparative analysis of the differential regulatory mechanisms of glucose and sucrose in cucumber under salt stress, elucidating the molecular mechanisms through which exogenous glucose and sucrose synergistically enhance salt tolerance in cucumber via multi-pathway interactions (Figure 11). The key findings include the following. (1) Both sugars significantly alleviate salt-induced growth inhibition, with mechanisms involving the maintenance of chlorophyll content, enhancement of photosynthetic efficiency, and improvement of osmotic adjustment capacity. (2) In terms of antioxidant defense, glucose predominantly activates the APX and RbohD pathways, whereas sucrose primarily modulates the POD system. (3) Transcriptomic analysis reveals that salt stress strongly induces the expression of key transcription factor families, including ERF, MYB, and NAC, and exogenous sugars enhance salt tolerance through differential regulation of these transcriptional networks. (4) Hormone signaling pathway analysis demonstrates that both glucose and sucrose facilitate the recovery of auxin, ET, and ABA signaling pathways, thereby mitigating the inhibitory effects of salt stress on plant growth and improving seedling salt tolerance. This study provides a solid theoretical foundation for understanding the role of sugar signaling in enhancing crop resistance to abiotic stress.

## Figures and Tables

**Figure 1 cimb-47-00754-f001:**
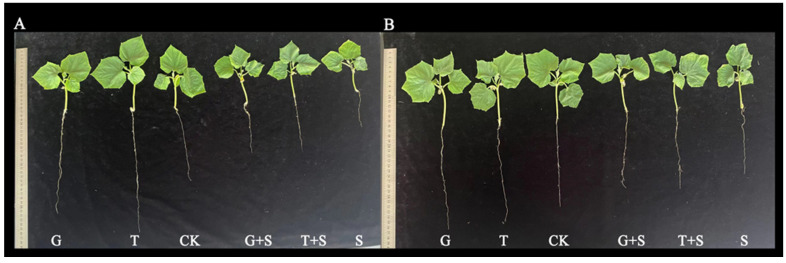
The growth phenotypes of cucumber seedlings under salt stress with exogenous sugars. Note: After different pretreatments, the seedlings were subjected to salt stress (150 mmol·L^−1^ NaCl) or non-salt stress for 6 days. CK: control; S: salt treatment; G: pretreated with 0.5 mmol·L^−1^ Glc; T: pretreated with 0.5 mmol·L^−1^ Suc; G + S: salt treatment after pretreatment with 0.5 mmol·L^−1^ Glc; T + S: salt treatment after pretreatment with 0.5 mmol·L^−1^ Suc. (**A**): Jinyou 1; (**B**): Xintaimici.

**Figure 2 cimb-47-00754-f002:**
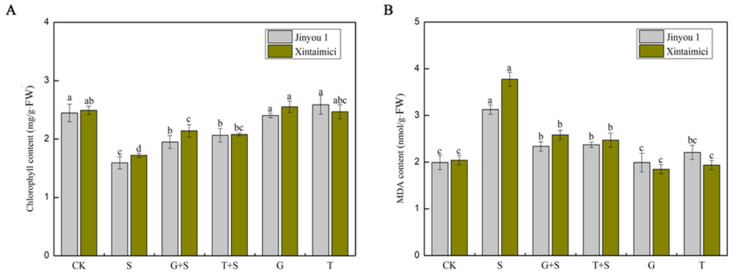
The detection of content of permeation regulating substances in cucumber leaves under salt stress with exogenous sugar. (**A**) Shows the contents of chlorophyll. (**B**) Shows the contents of malondialdehyde (MDA). Note: Values are means ± SD, n = 3; different letters indicate significant differences (*p* < 0.05).

**Figure 3 cimb-47-00754-f003:**
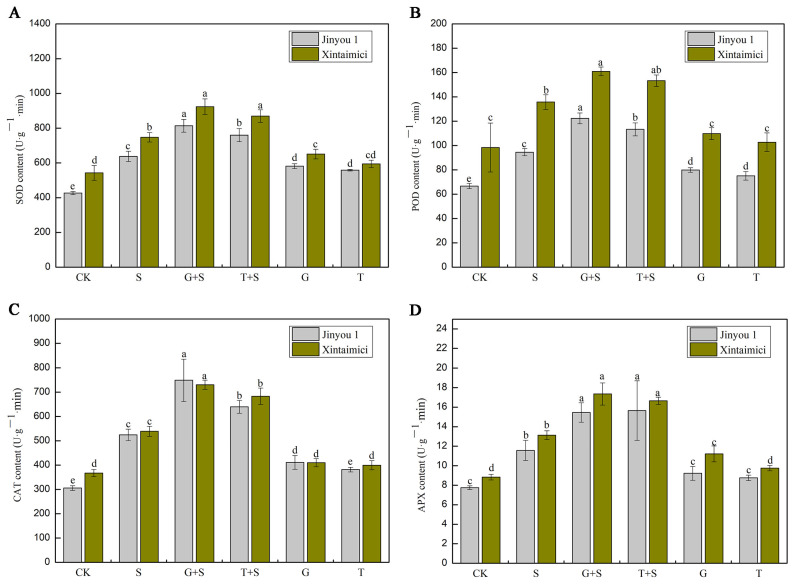
Effects of exogenous sugar application alleviates salt stress by scavenging ROS accumulation in leaves of cucumber seedlings. (**A**) Shows the contents of superoxide (SOD). (**B**) Shows the contents of peroxidase (POD). (**C**) Shows the contents of catalase (CAT). (**D**) Shows the contents of ascorbate peroxidase (APX). Note: Values are means ± SD (*n* = 3); different letters indicate significant differences (*p* < 0.05).

**Figure 4 cimb-47-00754-f004:**
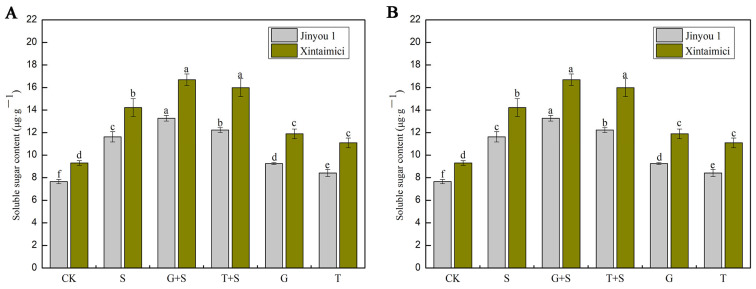
Effects of exogenous sugar on the content of osmotic adjustment substances in leaves of cucumber seedlings under salt stress. (**A**) Shows the contents of soluble sugar. (**B**) Shows the contents of soluble protein. Note: Values are means ± SD, n = 3; different letters indicate significant differences (*p* < 0.05).

**Figure 5 cimb-47-00754-f005:**
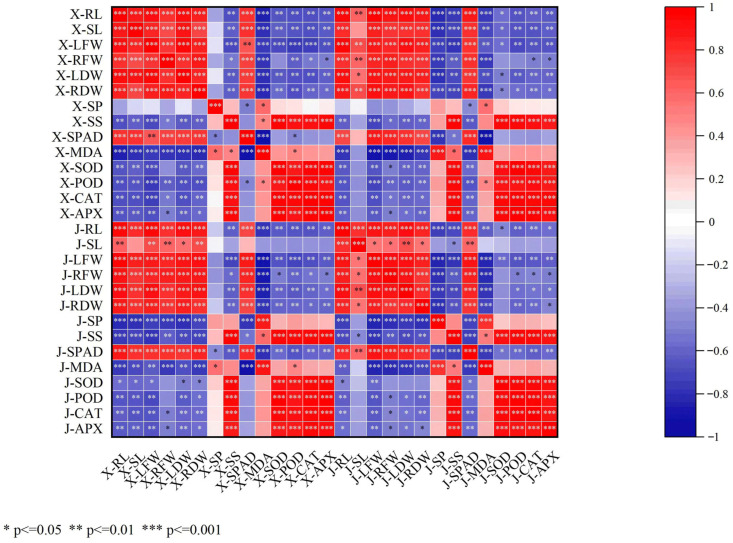
Correlation analysis matrix for each indicator. Note: X for Xintaimici. Root length for X-RL; stem length for X-SL; fresh weight of leaves for X-LFW; fresh weight of roots for X-RFW; dry weight of leaves for X-LDW; dry weight of roots for X-RDW; soluble protein for X-SP; soluble sugar for X-SS; relative chlorophyll content for X-SPAD; malondialdehyde for X-MDA; peroxidase for X-POD; catalase for X-CAT; superoxide dismutase for X-SOD; ascorbate peroxidase for X-APX. J for Jinyou1. Root length for J-RL; stem length for J-SL; fresh weight of leaves for J-LFW; fresh weight of roots for J-RFW; dry weight of leaves for J-LDW; dry weight of roots for J-RDW; soluble protein for J-SP; soluble sugar for J-SS; relative chlorophyll content for J-SPAD; malondialdehyde for J-MDA; peroxidase for J-POD; catalase for J-CAT; superoxide dismutase for J-SOD; ascorbate peroxidase for J-APX. * *p* ≤ 0.05, ** *p* ≤ 0.01, *** *p* ≤ 0.001. Red tones indicate positive correlations (positive correlation coefficients ranging from light red to dark red, corresponding to correlation coefficients from 0 to 1, with darker shades representing stronger positive correlations). Blue tones indicate negative correlations (negative correlation coefficients ranging from light blue to dark blue, corresponding to correlation coefficients from 0 to −1, with darker shades representing stronger negative correlations). White represents the absence of a statistically significant correlation (correlation coefficient close to 0 or non-significant correlation).

**Figure 6 cimb-47-00754-f006:**
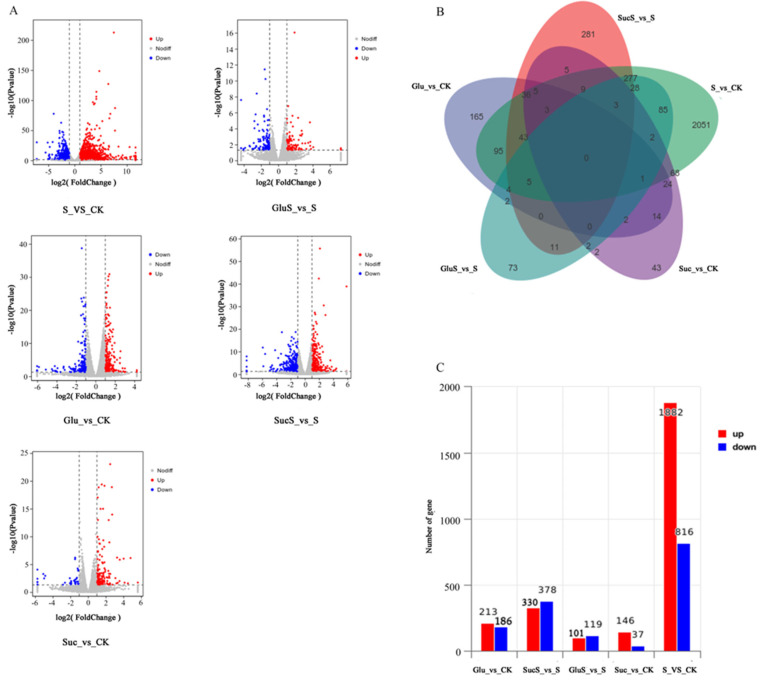
Transcriptional variation of cucumber plants under different treatments. (**A**) Volcano plots analysis of the DEGs in different treatments. (**B**) Venn diagrams showed the proportions of the up- and downregulated genes in three treatments. (**C**) The number of up- and downregulated genes in different treatments.

**Figure 7 cimb-47-00754-f007:**
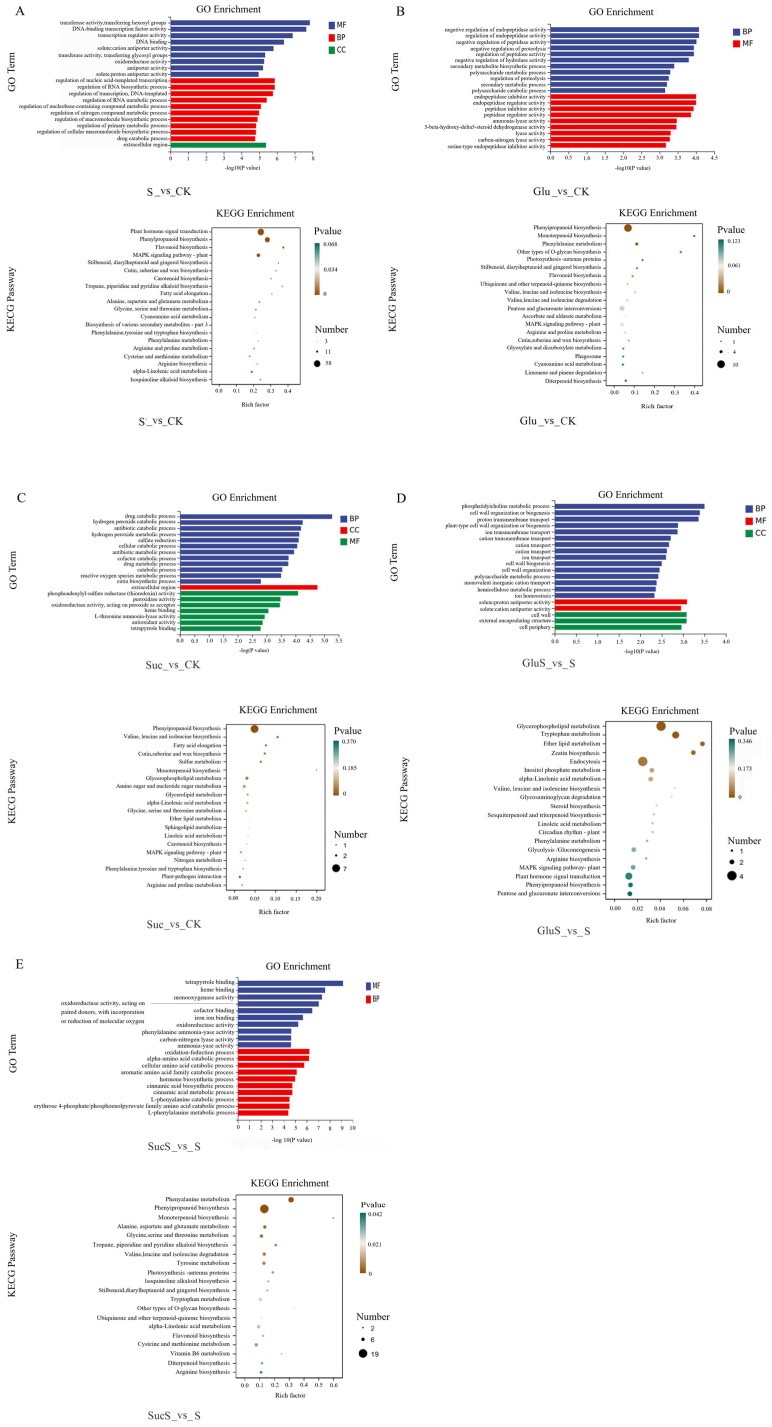
Enrichment analysis of transcriptional variation in cucumber plants under different treatments. (**A**) GO enrichment and KEGG analysis in S_vs_CK. (**B**) GO enrichment and KEGG analysis in Glu_vs_CK. (**C**) GO enrichment and KEGG analysis in Suc_vs_CK. (**D**) GO enrichment and KEGG analysis in GluS_vs_S. (**E**) GO enrichment and KEGG analysis in SucS_vs_S.

**Figure 8 cimb-47-00754-f008:**
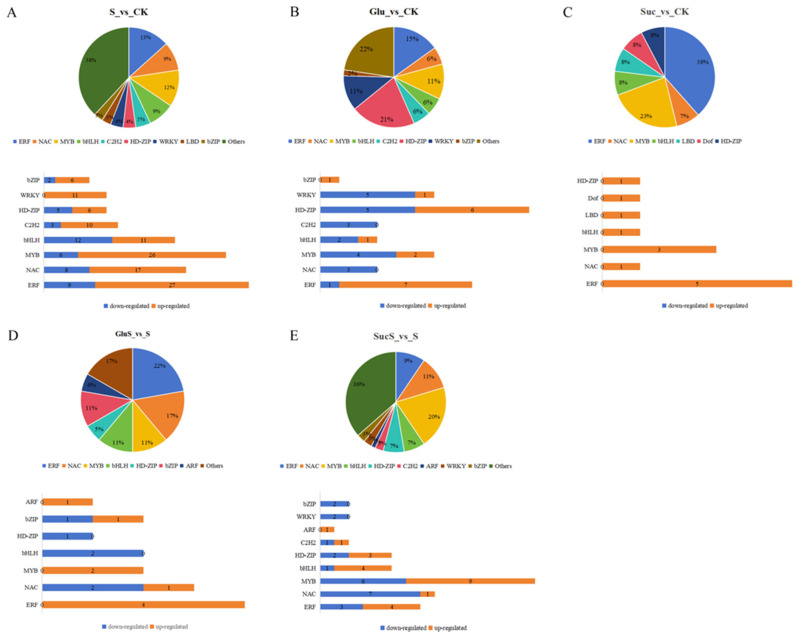
Distribution of transcription factor gene families expressed in different comparison groups. The pie chart shows the proportion of transcription factors in different comparison groups among the total transcription factors, while the bar chart presents the number of up-regulated and down-regulated transcription factor genes in the main comparison groups. The results of (**A**) S_vs_CK, (**B**) Glu_vs_CK, (**C**) Suc_vs_CK, (**D**) GluS_vs_S and (**E**) SucS_vs_S are respectively displayed in the figure.

**Figure 9 cimb-47-00754-f009:**
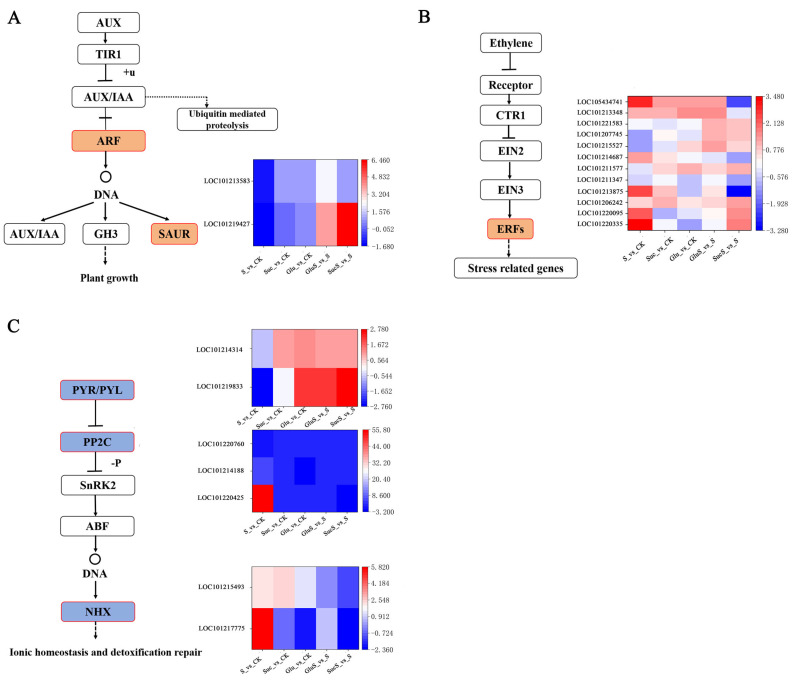
Effect of exogenous sugar on the expression of genes involved in hormone metabolism in cucumber seedlings under salt stress. (**A**) Effect of exogenous sugar on the expression of AUX-related genes under salt stress. (**B**) Effect of exogenous sugar on the expression of ET-related genes under salt stress. (**C**) Effect of exogenous sugar on the expression of ABA-related genes under salt stress.

**Figure 10 cimb-47-00754-f010:**
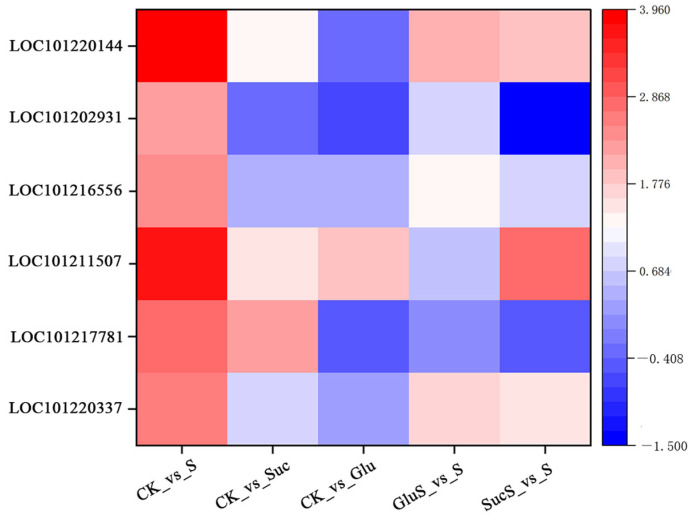
Effect of exogenous sugar on the expression of antioxidant enzyme-related genes under salt stress.

**Figure 11 cimb-47-00754-f011:**
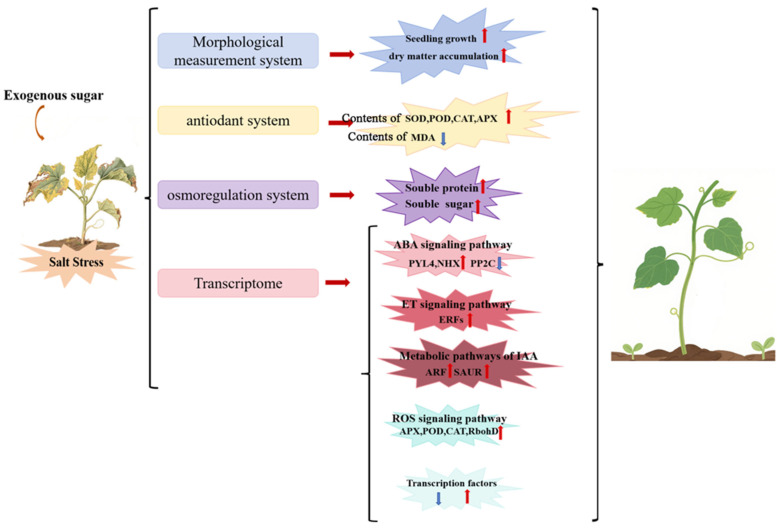
Schematic representation of the strategy of exogenous sugar to alleviate salt stress in cucumber seedlings. Red arrows denote an increase in the content of physiological indicators or up-regulated gene expression, whereas blue arrows denote a decrease in the content of physiological indicators or down-regulated gene expression.

**Table 1 cimb-47-00754-t001:** Effects of exogenous sugars on growth phenotypes of Jinyou 1 cucumber seedlings under salt stress.

Species	Concentration (mmol/L)	Root Length (cm/plant)	Stem Length (cm/plant)	Fresh Weight of Leaves (g/plant)	Fresh Weight of Roots (g/plant)	Dry Weight of Leaves (g/plant)	Dry Weight of Roots (g/plant)
Jinyou 1	CK	20.43 ± 1.4 a	5.16 ± 0.05 bc	10.29 ± 0.27 a	1.08 ± 0.21 ab	0.82 ± 0.05 b	0.05 ± 0.01 ab
	S	11.5 ± 1 c	5.09 ± 0.08 c	6.17 ± 0.05 c	0.4 ± 0.07 d	0.47 ± 0.03 d	0.02 ± 0.01 c
	T	25.77 ± 0.93 bc	5.59 ± 0.09 b	10.66 ± 0.31 a	1.25 ± 0.14 a	0.92 ± 0.08 a	0.06 ± 0.01 a
	G	31.5 ± 1 bc	8.1 ± 0.53 a	10.97 ± 0.64 a	1.31 ± 0.11 a	1.01 ± 0.06 a	0.06 ± 0.01 a
	G + S	15.43 ± 1.29 b	5.13 ± 0.15 bc	7.98 ± 0.29 b	0.88 ± 0.05 bc	0.68 ± 0.02 c	0.04 ± 0.01 bc
	T + S	15.67 ± 0.91 bc	5.2 ± 0.1 bc	8.07 ± 0.28 b	0.75 ± 0.04 c	0.66 ± 0.03 c	0.03 ± 0.01 c

Notes: Values are means ± SD (n = 3). Letters indicate significant differences at *p* < 0.05 according to Duncan’s multiple range tests.

**Table 2 cimb-47-00754-t002:** Effects of exogenous sugars on growth phenotypes of Xintaimici cucumber seedlings under salt stress.

Species	Concentration (mmol/L)	Root Length (cm/plant)	Stem Length (cm/plant)	Fresh Weight of Leaves (g/plant)	Fresh Weight of Roots (g/plant)	Dry Weight of Leaves (g/plant)	Dry Weight of Roots (g/plant)
Xintaimici	CK	19.5 ± 1 b	6.63 ± 0.47 b	8.02 ± 0.42 b	0.94 ± 0.14 ab	0.89 ± 0.05 b	0.08 ± 0.01 bc
	S	11.19 ± 1.01 d	4.93 ± 0.15 c	5.34 ± 0.21 c	0.53 ± 0.03 c	0.55 ± 0.04 d	0.03 ± 0.01 e
	T	21.5 ± 1 ab	7.69 ± 0.51 a	9.97 ± 0.42 a	1.04 ± 0.16 ab	1.26 ± 0.02 a	0.09 ± 0.01 b
	G	22.5 ± 1 a	6.9 ± 0.46 b	10.39 ± 0.52 a	1.15 ± 0.27 a	1.16 ± 0.1 a	0.11 ± 0.02 a
	G + S	15.2 ± 1 c	5.33 ± 0.42 c	6.09 ± 0.43 c	0.78 ± 0.08 bc	0.69 ± 0.03 c	0.05 ± 0.01 d
	T + S	14.1 ± 1 c	5.7 ± 0.1 c	5.73 ± 0.33 c	0.77 ± 0.09 bc	0.67 ± 0.03 c	0.06 ± 0.01 cd

Notes: Values are means ± SD (n = 3). Letters indicate significant differences at *p* < 0.05 according to Duncan’s multiple range tests.

**Table 3 cimb-47-00754-t003:** Summary of sequencing data for cucumber leaf samples.

Sample	Trimmed_Read_Numbe	Trimmed_Bases	Useful_Read%	Useful_Bases%
CK1	41801562	6301175388	98.78	98.61
CK2	38540682	5809769571	98.67	98.5
CK3	40756362	6144984555	98.56	98.42
Glu1	40849056	6157622624	98.57	98.4
Glu2	44114928	6643346865	98.66	98.39
Glu3	39828620	5996395865	98.68	98.39
GluSalt1	39724888	5986386299	98.71	98.51
GluSalt2	45010458	6776217132	98.55	98.25
GluSalt3	42776240	6444662183	98.63	98.4
Salt1	45333936	6823398963	98.88	98.56
Salt2	41571160	6259915355	98.72	98.45
Salt3	45697498	6889453658	98.66	98.5
Summary	758448556	76233328458		

## Data Availability

The research data used in the article has been provided in the Appendix A.

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
