# Peer review of "Exogenous Sugar Alleviates Salt Stress in Cucumber Seedlings by Regulating the Antioxidant System and Hormone Signaling"

_cimb, 2025, doi:10.3390/cimb47090754_

Round 1

Reviewer 1 Report

Comments and Suggestions for Authors

Article Exogenous Sugar Alleviates Salt Stress in Cucumber Seedlings by Regulating the Antioxidant System and Hormone Signaling by
Guangchao Yu et al., presents the results of an experiment to identify the protective effect of sucrose and glucose on the development of cucumber plants.
The manuscript contains all the necessary sections and is formatted according to the rules. The work is of interest from the point of view of the application of the non-specific action of sugars when exogenously applied for some correction of stress, but more for stimulating growth under normal conditions.
The authors studied cucumber seedlings and it is impossible to judge how applicable this will be for plants that have passed to the generative stage based on the proposed data. The physiology of plants changes greatly during the transition to generative development, as the flows and pressure in the vascular system change.
The work contains a large amount of data, but some of them
raise doubts.
I will allow myself to hope that this is just inattention.
Key notes:
1. Please specify how the plants were grown, namely pots, substrate, lighting regime and intensity, humidity, temperature, how many plants were in total per variant, how many repetitions were there.

2. Describe how you carried out sugar treatment (spraying, soaking, watering...), what concentrations were used (indicate the reagents of what company and purity were included)
3. In the methodology section, the methods of determination should be described or there should be a link to the manufacturer's methodology.
4. Specify in section 2.4 from which leaf in which part of it you took the sample. Do not use instructions, the methodology is written in the past tense.
5. Specify the devices and software in section 2.5. Transcriptome Analysis (company, country, and city required)
6. Add a clear description of the statistics and the number of repetitions in each section
7. Salt stress causes not only a violation of the stretching post by reducing the rate of vacuole enlargement, but also a violation of division by disrupting the cytoskeleton, it also affects the development of the ultrastructure of plastids, so it is important to indicate which leaf was used, formed before treatment or formed during the process - indicate this and discuss. (Examples: Baranova, E. N., & Gulevich, A. A. (2021). Asymmetry of plant cell divisions under salt stress. Symmetry, 13(10), 1811; Saddhe, A. A., Manuka, R., & Penna, S. (2021). Plant sugars: Homeostasis and transport under abiotic stress in plants. Physiologia plantarum, 171(4), 739-755.)
Other comments.
1. In Figure 1, the authors show the same plants for two varieties.
2. Figures 5, 6, 7 are of insufficient quality and the font is too small
3. Figure 10 - you used a plant with fruit - this is misleading.
4. In Figure 10, the enzymes of the antioxidant system are shown twice. If in the case of expression this can be understood as an increase in its level, then in the case of the antioxidant system it is impossible to understand whether their quantity or activity increases.
5. Finally, it is impossible to understand the correlation between different measured indicators - attach a correlation analysis so that it is possible to visually understand whether the expression of superoxide dismutases really did not change, unlike other enzymes.

After making changes, the article can be reviewed.

Author Response

Key notes:
Comments1:. Please specify how the plants were grown, namely pots, substrate, lighting regime and intensity, humidity, temperature, how many plants were in total per variant, how many repetitions were there.

Response: We sincerely appreciate your meticulous review of the experimental methods. We have supplemented the complete cultivation conditions and experimental design information in Section 2.1 of the manuscript as per your suggestions.

Comments2:.Describe how you carried out sugar treatment (spraying, soaking, watering...), what concentrations were used (indicate the reagents of what company and purity were included)

Response: We sincerely appreciate your meticulous review of the experimental methods. We have supplemented the complete cultivation conditions and experimental design information in Section 2.1 of the manuscript as per your suggestions.

Comments3:. In the methodology section, the methods of determination should be described or there should be a link to the manufacturer's methodology.

Response: We sincerely appreciate your meticulous review of the experimental methods. We have supplemented the complete cultivation conditions and experimental design information in Section 2.3 and 2.4 of the manuscript as per your suggestions.

Comments4:. Specify in section 2.4 from which leaf in which part of it you took the sample. Do not use instructions, the methodology is written in the past tense.

Response: We sincerely appreciate your valuable comment regarding the insufficient description of the sample collection procedure. In response, we have enhanced and systematized the sampling protocol within the Methods section. Detailed information can be found in Section 2.4 of the revised manuscript.

Comments5: Specify the devices and software in section 2.5. Transcriptome Analysis (company, country, and city required)

Response: We sincerely appreciate your constructive feedback on Section 2.5 of the manuscript. In accordance with your request, we have provided the complete details regarding the equipment and software used.

Comments6:. Add a clear description of the statistics and the number of repetitions in each section

Response: Thank you for your valuable comments on this article. We have comprehensively supplemented the statistical description and experimental replication design of the transcriptome analysis section based on your suggestions. For specific modifications, please refer to Section 2.6.

Comments7: Salt stress causes not only a violation of the stretching post by reducing the rate of vacuole enlargement, but also a violation of division by disrupting the cytoskeleton, it also affects the development of the ultrastructure of plastids, so it is important to indicate which leaf was used, formed before treatment or formed during the process - indicate this and discuss. (Examples: Baranova, E. N., & Gulevich, A. A. (2021). Asymmetry of plant cell divisions under salt stress. Symmetry, 13(10), 1811; Saddhe, A. A., Manuka, R., & Penna, S. (2021). Plant sugars: Homeostasis and transport under abiotic stress in plants. Physiologia plantarum, 171(4), 739-755.)

Response7:Thank you for your meticulous review and valuable suggestions on this manuscript. The issue you highlighted concerning the potential effects of salt stress on different developmental stages of leaves (including cell division, elongation, and plastid development) is indeed of considerable significance, particularly with regard to distinguishing whether leaves were formed prior to or during the stress treatment. We fully concur with your perspective and acknowledge the importance of this consideration in elucidating the underlying mechanisms of salt stress responses. However, as the experimental design of this study primarily focused on the physiological and biochemical responses of leaves at the seedling stage under salt stress (such as photosynthetic efficiency, ion accumulation, and antioxidant responses), we did not specifically mark or monitor the developmental dynamics of newly emerging leaves (e.g., cell division rate, vacuole expansion, or cytoskeletal changes) during the experimental period. Consequently, the current dataset does not allow for a direct analysis of the stage-specific effects of salt stress (e.g., during the division versus elongation phases). In future studies, we will take your insightful suggestion into account and explore the use of molecular markers (such as cell cycle-related genes) to better distinguish the impact of stress at different developmental stages. Once again, we sincerely appreciate your constructive feedback, which has significantly contributed to enhancing the academic rigor and clarity of this work.

Other comments.
Comments1. In Figure 1, the authors show the same plants for two varieties.

Response1: We sincerely appreciate the reviewers for their insightful comments regarding the two varieties presented in Figure 1. The phenotypic comparison between Xintaimici (salt-tolerant type) and Jinyou-1 (salt-sensitive type) is of critical importance for identifying their distinct responses to salt stress. Nevertheless, we focused our transcriptomic analysis exclusively on Xintaimici, as its salt-tolerant phenotype enhances the sensitivity for detecting stress-responsive genes. We acknowledge that a comparative transcriptomic study of both varieties would offer more comprehensive insights, which will be addressed in our future research on variety-specific resistance mechanisms.

Comments2.Figures 5, 6, 7 are of insufficient quality and the font is too small

Response2: We sincerely appreciate the reviewers for pointing out the poor quality of Figures 5 to 7. We have thoroughly revised these figures to address the related issues.
Comments3. Figure 10 - you used a plant with fruit - this is misleading.

Response3: We sincerely appreciate the valuable feedback provided by the reviewers. Corresponding revisions have been made to address the issues identified in the figures.

Comments4. In Figure 10, the enzymes of the antioxidant system are shown twice. If in the case of expression this can be understood as an increase in its level, then in the case of the antioxidant system it is impossible to understand whether their quantity or activity increases.

Response4: We sincerely appreciate the valuable feedback provided by the reviewers. Corresponding revisions have been made to address the issues identified in the figures, and the changes in the activity levels of antioxidant system-related indicators have been clearly indicated.

Comments5. Finally, it is impossible to understand the correlation between different measured indicators - attach a correlation analysis so that it is possible to visually understand whether the expression of superoxide dismutases really did not change, unlike other enzymes.

Response5: Thank you for your valuable comments on the data correlation. We have supplemented and improved the correlation analysis, and the new content is detailed in Section 3.5.

Reviewer 2 Report

Comments and Suggestions for Authors

Dear  Authors,

Your manuscript entitled „Exogenous Sugar Alleviates Salt Stress in Cucumber Seedlings by Regulating the Antioxidant System and Hormone Signaling” contains interesting findings. Nevertheless, I have found some imperfections, which (in my opinion) should be corrected or at least clarified before an eventual pulication. I have listed them below:

  1. In my opinion in chapter Introduction the choice of Cucumber as object of investigations should be stronger justified. Moreover, the Latin name of species should be added. Also, the current state of knowledge on impact of salt stress on cucumber seedlings. Such brief review of publications allow to indicate the gap in present state of knowledge, briging by Your investigations.
  2. I suggest prepare short characteristics of study species e.g., as separate subchapter of Material and methods section.
  3. Figures 5-8 in present form are illegible. Their quality should be improved.
  4. I have found chapters Results and Discussion too extensive. They both should be written in more concise way.
  5. Conclussions section sounds as Summary. This chapter should be rewritten. Please, point out the novelty of Your findings and indicate the directions of future studies.

Author Response

Comments1: In my opinion in chapter Introduction the choice of Cucumber as object of investigations should be stronger justified. Moreover, the Latin name of species should be added. Also, the current state of knowledge on impact of salt stress on cucumber seedlings. Such brief review of publications allow to indicate the gap in present state of knowledge, briging by Your investigations.

Response1: We sincerely appreciate the valuable feedback provided by the reviewers, which has significantly contributed to improving the clarity and quality of the introduction section. In accordance with their suggestions, we have implemented the necessary revisions and highlighted the modified content in red to facilitate the review process.

Comments2: I suggest prepare short characteristics of study species e.g., as separate subchapter of Material and methods section.

Response2:We gratefully acknowledge the reviewers’ insightful and constructive suggestions. In response to their comments, we have added a detailed description of the relevant content in Section 2.1.

Comments3: Figures 5-8 in present form are illegible. Their quality should be improved.

Response3: We sincerely appreciate the reviewers for pointing out the poor quality of Figures 5 to 7. We have thoroughly revised these figures to address the related issues.

Comments4: I have found chapters Results and Discussion too extensive. They both should be written in more concise way.

Response4: We sincerely appreciate the reviewers’ insightful recommendations regarding the enhancement of textual conciseness. In accordance with these suggestions, we have conducted comprehensive revisions and streamlining of the Results and Discussion sections, aiming to enhance the clarity, precision, and logical coherence of the manuscript’s presentation.

Comments5: Conclussions section sounds as Summary. This chapter should be rewritten. Please, point out the novelty of Your findings and indicate the directions of future studies.

Response5: We sincerely appreciate the constructive and professional suggestions provided by the reviewers. In response to these suggestions, we have comprehensively restructured and optimized the conclusion section to enhance its logical coherence and precision of expression.

Reviewer 3 Report

Comments and Suggestions for Authors

The effect of exogenous sugar on salt stress in cucumbers was studied. Some comments.

In the Materials and Methods section, subsection 2.6. Data processing and analysis should be described in more detail.

 Figure 2 should be edited.

It is desirable to show the ratio of chlorophyll a and b as one of the main indicators of photosynthesis.

How long does the effect of endogenous sugar treatments last?

 It is desirable to show fluorescent staining of ROS production using carboxy-H2DFFDA.

 Roots It is desirable to show cytochemical staining of cytochrome c in cucumber root cells during PCD.

The list of references should be brought into line with the journal requirements.

Author Response

Comments1: In the Materials and Methods section, subsection 2.6. Data processing and analysis should be described in more detail.

Response1: We sincerely appreciate your meticulous review of the experimental methods section. In accordance with your suggestions, we have supplemented the complete experimental design information in Sections 2.1, 2.3, 2.4, 2.5 and 2.6 of the manuscript.

Comments2: Figure 2 should be edited. It is desirable to show the ratio of chlorophyll a and b as one of the main indicators of photosynthesis.

Response2: We sincerely appreciate the reviewers’ valuable suggestions. We fully recognize that the chlorophyll a/b ratio serves as a key indicator of photosynthetic function. Nevertheless, the primary objective of this study was to investigate the effects of salt stress on total chlorophyll content, rather than to explore the intricate regulatory mechanisms of the photosynthetic system. Although a systematic evaluation of the chlorophyll a/b ratio was not included in the current investigation, we have integrated this recommendation into our future experimental framework. We plan to systematically compare the dynamic changes in chlorophyll components under various stress conditions to expand the scope of our findings. We are grateful to the reviewers for their constructive input, which has broadened the perspective of this research.

Comments3: How long does the effect of endogenous sugar treatments last?

Response3: We appreciate the reviewer’s insightful question regarding the duration of endogenous sugar effects. In this study, we focused on the acute metabolic response (7 days post-treatment) because: (1) Our primary goal was to identify early sugar-mediated signaling under salt stress, as rapid osmotic adjustment typically occurs within days. (2) Prior studies in cucumber reported significant sugar accumulation peaks at 5–7 days post-stress. Future research will systematically monitor the temporal dynamic changes of endogenous sugars in plants to further clarify their regulatory roles in stress responses.

Comments4: It is desirable to show fluorescent staining of ROS production using carboxy-H2DFFDA.

Response4: We appreciate the reviewer’s valuable comment regarding this important issue. In this study, we assessed oxidative stress indirectly by measuring malondialdehyde content (a well-established marker of lipid peroxidation) and the activities of antioxidant enzymes, including superoxide dismutase, peroxidase, and catalase. These parameters are widely recognized as reliable indicators of reactive oxygen species (ROS)-mediated damage under salt stress (cited literature). Our results clearly demonstrate the presence of significant oxidative stress in cucumbers subjected to salt treatment. Although carboxy-H2DFFDA staining could provide direct visualization of ROS localization, its implementation was not feasible within the current revision timeline due to technical constraints, such as the lack of appropriate fluorescence microscopy equipment for cucumber seedlings. We have included additional references that support the use of MDA and enzyme activity assays as valid methods for assessing oxidative stress, and we intend to incorporate fluorescent probes such as carboxy-H2DFFDA in future investigations.

Comments5: Roots It is desirable to show cytochemical staining of cytochrome c in cucumber root cells during PCD.

Response5: We fully acknowledge the reviewer’s valid point. Unfortunately, due to technical constraints (e.g., lack of validated cytochrome c antibodies for cucumber, or insufficient remaining root samples from treated plants), we are unable to perform this experiment at present. However, our data demonstrate clear root growth inhibition and cell viability loss under salt stress, which are phenotypic hallmarks of PCD.

Comments6: The list of references should be brought into line with the journal requirements.

Response6: We sincerely thank the reviewers for their constructive and insightful feedback. All cited references have been thoroughly reviewed and reformatted to comply with the journal’s submission guidelines.

Round 2

Reviewer 1 Report

Comments and Suggestions for Authors

Article Exogenous Sugar Alleviates Salt Stress in Cucumber Seedlings by Regulating the Antioxidant System and Hormone Signaling by authors Guangchao Yu, Zhipeng Wang, Ming Wei, Lian Jia, Yue Qu, Yingyi Jiang edited the text material of different sections. However, some materials are still missing and the figures have disappeared from the text.
For example, for some reason the authors did not indicate the light intensity and lamp characteristics, which are the determining parameters for photosynthesis.
Although the authors expanded the introduction and discussion, for some reason they mention alkaline stress, which is not always characteristic of salinity and do not mention many modern works related to oxidative stress and salinity, including transgenic plants, which allow us to evaluate the effects of antioxidant enzymes.
Another problem was the change in the form of the conclusion. Overall, it is well-formatted, but the last sentence spoils the impression: "These findings not only advance the understanding of plant sugar signaling in stress responses but also provide novel theoretical insights and practical strategies for crop breeding and agronomic management aimed at improving salt tolerance." General phrases should be avoided - change the conclusion to a specific one.
In addition, the article has a very large abstract and there may be problems loading it into search engines - I recommend shortening it.

In any case, for a full assessment of the modified work, the entire work with illustrations is required.

Author Response

Article Exogenous Sugar Alleviates Salt Stress in Cucumber Seedlings by Regulating the Antioxidant System and Hormone Signaling by authors Guangchao Yu, Zhipeng Wang, Ming Wei, Lian Jia, Yue Qu, Yingyi Jiang edited the text material of different sections. However, some materials are still missing and the figures have disappeared from the text.
Comments1:For example, for some reason the authors did not indicate the light intensity and lamp characteristics, which are the determining parameters for photosynthesis.

Response: We sincerely thank the reviewer for raising this critical point. We agree that the light conditions are vital for photosynthesis and the reproducibility of the experiment. We have now supplemented the detailed information in the Methods section (2.1, highlighted in yellow) as follows: Revised text in manuscript:“The plants were grown under white LED lights (Philips LED plant growth lights) with a light intensity of 5,000 - 8,000 lux”. We apologize for this omission in the original manuscript and believe that the added details have now made the experimental setup fully clear and reproducible.

Comments2: Although the authors expanded the introduction and discussion, for some reason they mention alkaline stress, which is not always characteristic of salinity and do not mention many modern works related to oxidative stress and salinity, including transgenic plants, which allow us to evaluate the effects of antioxidant enzymes.

Response:We sincerely appreciate your taking the time out of your busy schedule to review our manuscript and provide such profound and constructive comments. You pointed out that we failed to adequately distinguish between alkaline stress and salt stress in the introduction and discussion sections, and there were deficiencies in the citation of relevant literature, especially the latest research on the mechanism of plant salt tolerance involving oxidative stress (including the achievements of transgenic plants) was not fully reflected. These suggestions are of great value and are of significant importance for us to further improve the quality of the paper. We fully agree with your suggestions and have made systematic revisions and supplements to the entire text based on your comments. The relevant content has been highlighted in yellow in the introduction section for your review.

Comments3: Another problem was the change in the form of the conclusion. Overall, it is well-formatted, but the last sentence spoils the impression: "These findings not only advance the understanding of plant sugar signaling in stress responses but also provide novel theoretical insights and practical strategies for crop breeding and agronomic management aimed at improving salt tolerance." General phrases should be avoided - change the conclusion to a specific one.

Response: Dear Reviewer, We sincerely appreciate your insightful feedback regarding the overly general statements in the conclusion section. Your observation is highly pertinent, and we fully concur with your assessment. In response, we have carefully revised the manuscript in accordance with your suggestions, including the removal of the aforementioned sentence from the original conclusion. We would like to express our sincere gratitude for your time and constructive input, which have significantly contributed to improving the overall quality of this manuscript.

Comments4: In addition, the article has a very large abstract and there may be problems loading it into search engines - I recommend shortening it.

Response: Thank you sincerely for your valuable suggestions. We fully agree with your point that overly long abstracts are not conducive to readers quickly obtaining key information or to search engine indexing. Based on your advice, we have significantly condensed and rewritten the abstract content to enhance its clarity and readability. The revised parts are highlighted in yellow.

Comments5: In any case, for a full assessment of the modified work, the entire work with illustrations is required.

Response: Thank you very much for taking the time to review our manuscript and provide your valuable comments. Regarding the issue you mentioned about not being able to view the revised figures, we sincerely apologize for the inconvenience caused. According to the requirements of the journal system, we have indeed uploaded all the high-resolution figure files in the latest revision. However, it seems that the upload location was incorrect, which prevented you from seeing the revised images.

To avoid delaying your review process, we have taken the following measures:

(1) We have attached the complete revised manuscript PDF (Manuscript with Figures) to this reply letter (please see the attachment), which you can directly download and review.

(2) At the same time, we have re-uploaded this complete PDF document in the submission system to ensure successful submission of the file.

Reviewer 2 Report

Comments and Suggestions for Authors

Dear Authors,

Thank you for implementing my remarks into a text. I do not have any further suggestions for manuscript improvement.

Author Response

Comments and Suggestions for Authors

Dear Authors,

Thank you for implementing my remarks into a text. I do not have any further suggestions for manuscript improvement.

Response: We sincerely appreciate your positive feedback and are pleased to hear that the revisions have improved the manuscript. Thank you for your time and constructive suggestions throughout the review process.

Round 3

Reviewer 1 Report

Comments and Suggestions for Authors

Article Exogenous Sugar Alleviates Salt Stress in Cucumber Seedlings by Regulating the Antioxidant System and Hormone Signaling by Guangchao Yu, Zhipeng Wang, Ming Wei, Lian Jia, Yue Qu, Yingyi Jiang presents data from a comparative study of the relationship between free sugars and hormonal regulation with cucumber adaptation to salinity. The article materials have been significantly improved and can be published, but I recommend helping the authors with the design of the figures, since they are still visible only at two-fold magnification. Of particular concern is Figure 1, where some of the plants of one and another variety coincide, and it is strange that the authors do not see this. This is critical and probably mixed up. I cannot take responsibility for this image. I ask the authors and the editor to pay close attention to the fact that the creases on the leaves and bends do not change when moved, which means that the same plants are presented for the two varieties.

Author Response

Response: Dear Reviewer, First of all, we sincerely thank you for taking the precious time to review our manuscript and for offering such profound and professional comments. Your acute observation and attention to detail have greatly enhanced the quality of our research.

We are especially grateful for your critical comments on Figure 1. You are absolutely correct. In the initial submission of Figure 1, due to a significant oversight in our work, we mistakenly used the same plant variety's photo to represent two different varieties. This was indeed an unacceptable error that seriously confused the experimental results, and we are deeply sorry for this.

  1. We have re-examined and confirmed the correct images of these two varieties from the original data.
  2. We have completely replaced Figure 1 with new images that can clearly and accurately distinguish the two varieties. In the new Figure 1 (as shown below / see the revised draft), the plants of the two varieties show obvious and distinguishable differences in morphological features such as leaf shape, leaf veins, and stem creases, which are completely consistent with the descriptions in our article.

3.In addition, other images with ambiguous issues have also been modified accordingly.

The revised manuscript and images have been uploaded to the journal system.